# FtsN maintains active septal cell wall synthesis by forming a processive complex with the septum-specific peptidoglycan synthases in *E. coli*

Zhixin Lyu [1], Atsushi Yahashiri[2], Xinxing Yang[1,3], Joshua W. McCausland[1], Gabriela M. Kaus[2], Ryan McQuillen [1], David S. Weiss[2] ✉ & Jie Xiao [1] ✉

FtsN plays an essential role in promoting the inward synthesis of septal peptidoglycan (sPG) by the FtsWI complex during bacterial cell division. How it achieves this role is unclear. Here we use single-molecule tracking to investigate FtsN's dynamics during sPG synthesis in *E. coli*. We show that septal FtsN molecules move processively at ~9 nm s⁻¹, the same as FtsWI molecules engaged in sPG synthesis (termed sPG-track), but much slower than the ~30 nm s⁻¹ speed of inactive FtsWI molecules coupled to FtsZ's treadmilling dynamics (termed FtsZ-track). Importantly, processive movement of FtsN is exclusively coupled to sPG synthesis and is required to maintain active sPG synthesis by FtsWI. Our findings indicate that FtsN is part of the FtsWI sPG synthesis complex, and that while FtsN is often described as a "trigger" for the initiation for cell wall constriction, it must remain part of the processive FtsWI complex to maintain sPG synthesis activity.

Most bacteria are completely encased in a peptidoglycan (PG) sacculus or cell wall that confers cell shape and protects against lysis by internal osmotic pressure, which can be as high as ~3 atm in Gram-negative *Escherichia coli*[1] and 20 atm in Gram-positive *Bacillus subtilis*[2]. The importance of the cell wall is underscored by the fact that it is one of the most successful antibiotic targets[3,4].

During cell division, bacteria must synthesize and remodel their protective cell wall to accommodate the splitting of a mother cell into two daughter cells[5]. Bacterial cell division is mediated by the divisome, a loosely-defined collection of proteins that form a contractile ring-like assemblage at the division site. In *E. coli* the divisome contains over 30 different types of proteins, of which ten are essential and considered to constitute the core of the division apparatus[6,7]. The ten essential division proteins are recruited to the divisome in a mostly sequential fashion, starting with the tubulin-like GTPase FtsZ[8,9] and ending with

the bitopic membrane protein FtsN, whose arrival coincides with the onset of visible constriction[10,11]. Other noteworthy divisome proteins include FtsA, which links FtsZ polymers to the membrane[12], the core septal PG (sPG) synthase complex composed of the polymerase FtsW and transpeptidase FtsI[13,14], and the FtsQLB complex, which regulates FtsWI activity[15,16] (also see reviews[5–7]). According to current models, FtsN acts through FtsA in the cytoplasm and the FtsQLB complex in the periplasm to activate synthesis of sPG by the FtsWI synthase complex[15–22].

Advanced high resolution and single-molecule imaging are providing important new insights into the organization of the divisome and the control of sPG synthesis[23–34]. One important finding is that FtsZ uses GTP hydrolysis to move around the septum by treadmilling[35–38], which is the apparent directional movement of a polymer caused by continuous polymerization at one end and depolymerization at the

[1]Department of Biophysics and Biophysical Chemistry, Johns Hopkins School of Medicine, Baltimore, MD 21205, USA. [2]Department of Microbiology and Immunology, University of Iowa Carver College of Medicine, Iowa City, IA 52242, USA. [3]The Chinese Academy of Sciences Key Laboratory of Innate Immunity and Chronic Disease, School of Basic Medical Sciences, Division of Life Sciences and Medicine, University of Science and Technology of China, Hefei, China. ✉e-mail: david-weiss@uiowa.edu; xiao@jhmi.edu

other end, with individual monomers in the middle remaining stationary. Furthermore, in both *E. coli* and *B. subtilis*, FtsZ's treadmilling dynamics drive directional movement of the sPG synthesis complex FtsWI at a speed of ~30 nm s$^{-1}$ [35,36], likely through a Brownian ratchet mechanism[39]. Thus, FtsZ uses its GTPase activity-dependent treadmilling dynamics to function as a linear motor to distribute sPG enzyme complexes along the septum to ensure a smooth, symmetric septum synthesis[23,35,36,39].

More recently, we discovered that the *E. coli* divisome contains a second population of FtsWI, one that moves processively at ~9 nm s$^{-1}$ [40]. Movement of this slower population is driven by active sPG synthesis (termed as on the sPG-track) rather than FtsZ treadmilling (termed as on the Z-track). Similar FtsZ-independent but sPG synthesis-dependent processive populations of FtsW and PBP2x were first observed in *S. pneumoniae*[38]. In *E. coli* individual FtsW or FtsI molecules can transition back-and-forth between the fast- and slow-moving populations. In cells depleted of FtsN, the active, slow-moving population of FtsWI on the sPG-track is diminished while the inactive, fast-moving population of FtsWI on the Z-track is enhanced. These findings imply that FtsN activates sPG synthesis, at least in part, by increasing the number of FtsWI molecules on the sPG-track[40], but it is unclear how this increase is achieved.

In this work, we use single-molecule imaging to investigate the organization and dynamics of FtsN at the septum and how they are coupled to sPG synthesis. Consistent with a previous report, we observed that FtsN exhibits distinct spatial organization and dynamics from the FtsZ-ring[41]. Most importantly, we report that single FtsN molecules at the septum of constricting cells move exclusively and processively at a slow speed of ~9 nm s$^{-1}$. The processive movement of FtsN depends on active sPG synthesis but not FtsZ's treadmilling dynamics. These dynamic behaviors are identical to those of the slow-moving, active population of FtsWI on the sPG-track. We also observed that the so-called "essential" (E) domain of FtsN, a helix bundle proposed to mediate the activation of FtsWI, is both required and sufficient for the processive movement of FtsN on the sPG-track. *In toto*, our findings indicate that FtsN is a member of the active FtsWI synthase complex on the sPG-track, that the association of FtsN with FtsWI is mediated by FtsN's E domain, and that FtsN must remain in the FtsWI complex to sustain processive synthesis of sPG.

## Results

### Construction of functional FtsN fusions

FtsN has at least four functional domains (Fig. 1a and Supplementary Fig. 1): an N-terminal cytoplasmic tail (FtsN$^{Cyto}$) that interacts with FtsA;[17,19,20] a transmembrane domain (FtsN$^{TM}$) that anchors FtsN to the inner membrane[42]; a periplasmic essential domain (FtsN$^{E}$) that is composed of three helices and responsible for activating sPG synthesis[11,43], and a C-terminal periplasmic SPOR domain (FtsN$^{SPOR}$) that binds to denuded PG glycan strands, which are transiently present at the septum during cell wall constriction[11,43–47]. To identify functional fluorescent fusions of FtsN for imaging experiments, we screened 11 FtsN fusions that have the green fluorescent protein mNeonGreen (mNG)[48] fused to the N-terminus, C-terminus or inserted at internal positions of FtsN (Supplementary Fig. 1). These fusions were expressed from plasmids in an FtsN-depletion background to test their functionality (Supplementary Fig. 2, Supplementary Table 4). We were able to identify an N-terminal and an internal (termed sandwich, between E60 and E61) fusion of FtsN that supported normal growth on solid and liquid media in FtsN depletion backgrounds (Supplementary Fig. 2A, B), and exhibited correct midcell localization during cell division (Supplementary Fig. 2C). Based on these results we constructed additional fusions to various fluorescent proteins for different imaging purposes, including the N-terminal fusions mEos3.2-FtsN and GFP-FtsN, and the sandwich fusion FtsN-Halo$^{SW}$ (Supplementary Fig. 3, Supplementary Fig. 4 and "Methods" section). These fusions were

cloned downstream of a synthetic isopropyl $\beta$-D-1-thiogalactopyranoside (IPTG)-inducible promoter and integrated into the chromosome at a phage attachment site in an FtsN-depletion strain, where *ftsN's* native promoter was replaced with the arabinose-dependent $P_{BAD}$ promoter[49] (Supplementary Table 1). Except where stated otherwise, all experiments described below used cells grown in M9-glucose minimal media supplemented with IPTG in the absence of arabinose. Under these conditions, the fluorescent FtsN fusion protein is the sole source of FtsN in the cells (Supplementary Fig. 4). Expression, stability and functionality of the fusions were validated by Western blotting and cell growth measurements (Supplementary Fig. 4 and Supplementary Table 4).

### The FtsN-ring exhibits different spatiotemporal organization and dynamics from the FtsZ-ring

FtsN is expressed at a level of ~300 molecules per cell based on ribosome-profiling[50], which we verified directly by quantitative Western blotting (Supplementary Fig. 5). Using Stimulated Emission Depletion (STED)[51] and Structured Illumination Microscopy (SIM)[52], a previous study reported that FtsN forms a discontinuous, ring-like structure (FtsN-ring) at the midcell, similar to—but physically distinct from—the FtsZ-ring[41]. To investigate the spatial organization of the FtsN-ring with a higher spatial resolution than SIM or STED, we used astigmatism-based three-dimensional (3D) single-molecule localization microscopy (SMLM)[53] to image an mEos3.2-FtsN fusion that was the sole source of FtsN in live *E. coli* cells (Strain EC4443 in Supplementary Table 1). We used this fusion because mEos3.2 is so far the best performing fluorescent protein for SMLM imaging[54].

We observed that FtsN-rings are patchy (Fig. 1b) as previously reported, and that FtsN exhibits significant membrane localization along the perimeter of the cell. The high spatial resolutions (~50 nm in *xy* and ~80 nm in *z*, Supplementary Fig. 6) revealed that FtsN-rings have a comparable width and thickness to FtsZ-rings[27,29,30,32,34] (Fig. 1b, Supplementary Fig. 6C, and Supplementary Table 5) and that FtsN molecules in the FtsN-ring are more homogenously distributed than FtsZ molecules in the FtsZ-ring, as indicated by the autocorrelation analysis (Fig. 1c). To explore how FtsN-rings assemble and disassemble during cell division, we calculated the midcell localization percentages of FtsN by dividing midcell ring fluorescence by the whole cell fluorescence. We observed that maximally ~20% of cellular FtsN molecules accumulated in the FtsN-ring (Fig. 1d). This value is in agreement with a recent study using a fluorescent *ftsN* fusion expressed from *ftsN's* native chromosomal locus[55]. In contrast, the FtsZ-ring contained up to ~45% of the cellular pool of FtsZ (Fig. 1d). When cells of different ring diameters were arranged to generate a pseudo time lapse representing the cell wall constriction process, we observed that FtsZ-rings assembled at a ring diameter of ~950 nm, whereas FtsN-rings were first visible at a diameter of ~600 nm. The apparent timing of ring disassembly also differed, as FtsZ-rings drastically diminished in fluorescence intensity starting at ~600 nm, whereas FtsN-rings disassembled only modestly through ~300 nm (Fig. 1d). These observations are consistent with previous evidence that FtsN- and FtsZ-rings have different spatiotemporal organizations[41].

To examine whether the FtsN-ring is a static or dynamic structure, we investigated the turnover of FtsN subunits in the ring using Fluorescence Recovery After Photobleaching (FRAP) of a GFP-FtsN fusion (Supplementary Fig. 7 and Supplementary Movie 1, Strain EC4240 in Supplementary Table 1). GFP is well-suited for this purpose because of its low photostability. We bleached half of the ring and observed that the recovery curve of GFP-FtsN exhibited two apparent phases (Fig. 1e, red), a fast phase with a recovery half time $\tau_{1/2} = 2.9 \pm 0.8$ s, and a slow phase with $\tau_{1/2} = 54 \pm 10$ s ($\mu \pm$ s.e.m., $n = 58$ cells). Most interestingly, we only observed a ~70% recovery of FtsN's intensity compared to that prior to bleaching, indicating that a subpopulation of FtsN molecules was stationary on the time scale of the experiment (150 s).

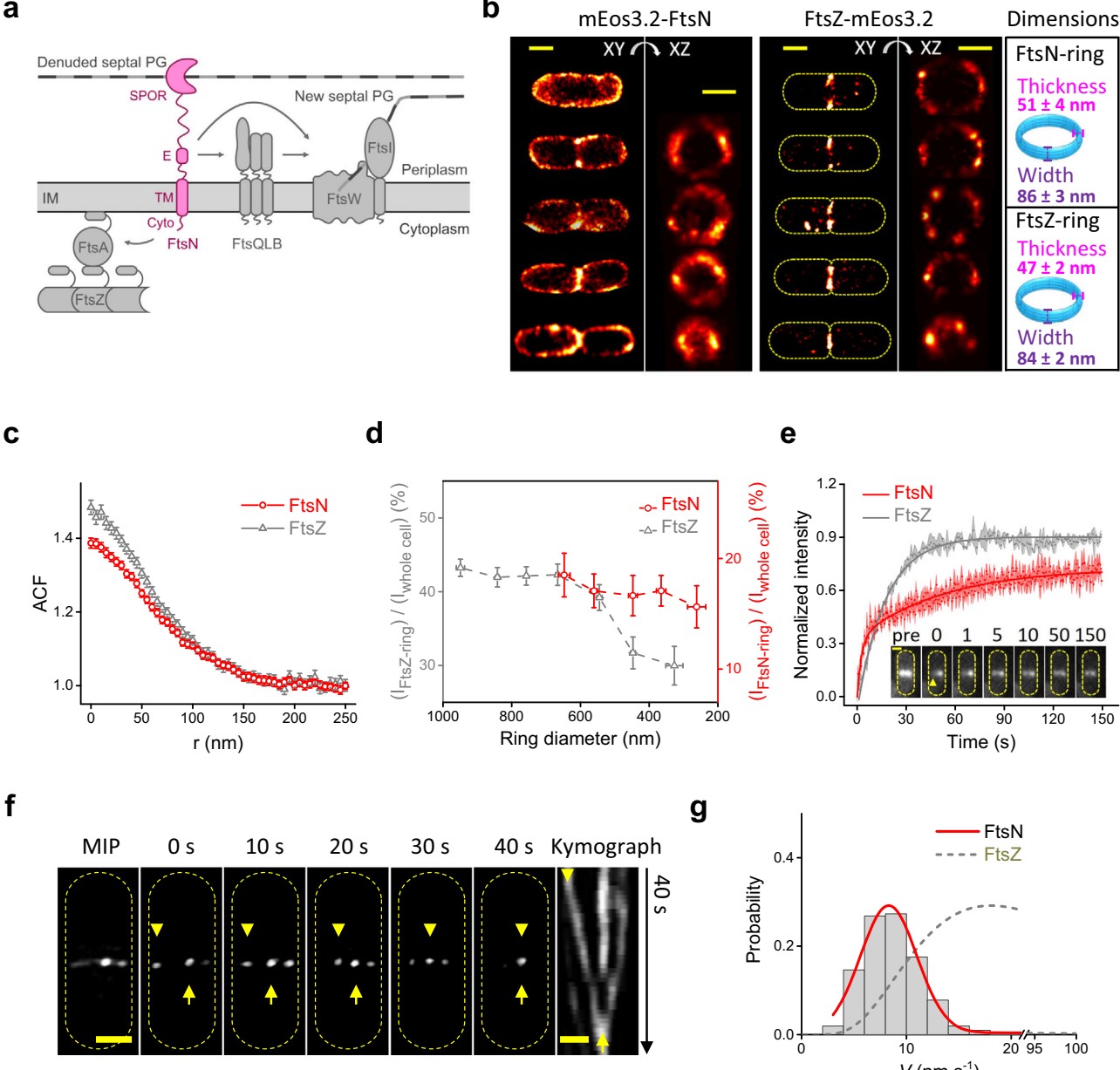

**Fig. 1 | FtsN-ring has a different organization and dynamics compared to FtsZ-ring. a** Schematic drawing of FtsN's domain organization and interactions with other divisome proteins. **b** Three-dimensional (3D) superresolution images show that FtsN-rings (left, mEos3.2-FtsN fusion, Strain EC4443 in Supplementary Table 1) are patchy but more homogenous than FtsZ-rings (middle, data from a previous work[30]). Yellow dashes mark cell outlines. Scale bars, 500 nm. Toroid ring models (cyan) with the average ring dimensions are shown on the right. **c** Mean spatial autocorrelation function (ACF) curve of FtsN-rings (red) averaged from all individual cells' ACFs along the circumference of the ring ($r$) has lower correlation values at short distances and longer characteristic decay length than FtsZ-ring's ACF (gray), indicating a more homogenous distribution of FtsN. **d** Pseudo time course of FtsN's midcell localization percentages ($I_{ring}/I_{whole\ cell}$) during cell division suggests that FtsN disassembles later than FtsZ (gray). In (**b**–**d**), $n = 72$ rings for FtsN and $n = 103$ rings for FtsZ (data from a previous work[30]). Data are presented as mean ±

s.e.m. **e** Mean FRAP recovery curve of FtsN (red, $n = 58$ cells, GFP-FtsN, Strain EC4240) exhibits slower and lower recovery than that of FtsZ (gray, data from a previous work[36]). Error shadow represents standard deviation. Examples of raw FRAP images are shown as inset (also see Supplementary Fig. 7a). Scale bar, 300 nm. **f** Maximum intensity projection (MIP, left), montages (0–40 s) from time lapse imaging of an mNG-FtsN fusion-expressing cell (Strain EC4564) and the corresponding kymograph (right) imaged using TIRF-SIM. Scale bar, 300 nm. Arrowhead and arrow: a moving cluster and a stationary cluster respectively. **g** Speed distribution of processively moving FtsN clusters combined from both TIRF-SIM and TIRF imaging (gray columns, $8.7 \pm 0.2$ nm s$^{-1}$, $\mu \pm$ s.e.m., $n = 205$ clusters) overlaid with the corresponding fit curve (red) and a fit curve of FtsZ's treadmilling speed distribution (dash gray, data from a previous work[40]). The $x$-axis breaks from 21 to 94 nm s$^{-1}$ to accommodate the distinct speed distributions between FtsN and FtsZ clusters. Source data are provided as a Source Data file.

In comparison, at the same time scale the FtsZ-ring recovered with a half time of ~16 s and to ~90% of the intensity prior to bleaching (Fig. 1e, gray, data from a previous work[36]). The fast recovery phase of FtsN is also previously observed by Söderström et al.[41], and is most likely due to the random diffusion of FtsN molecules in and out of the septum as expected for a typical inner membrane protein

(see "Methods" section). The slow recovery phase, however, has not been observed previously. It was not due to the global photobleaching nor photoblinking of GFP because no fluorescence recovery was observed in adjacent cells (Supplementary Fig. 7). The slow phase is significantly slower than that of FtsZ, indicating that the FtsN-ring exhibits a new type of dynamics compared to the FtsZ-ring.

## FtsN clusters exhibit slow, directional motions

To investigate what type of dynamics contribute to the observed slow FRAP behavior, we imaged FtsN-rings using an mNG-FtsN fusion (Strain EC4564 in Supplementary Table 1) with Structured Illumination Microscopy coupled with total internal reflection excitation (TIRF-SIM)[56,57]. We chose mNG for these experiments because of its enhanced brightness and photostability, which allowed us to monitor the dynamics of FtsN-rings with a spatial resolution of ~100 nm and a time resolution of 100 ms. Similar to what we observed in 3D-SMLM imaging, the fluorescence of FtsN-rings was patchy and clustered (Fig. 1f and Supplementary Fig. 8B). Kymograph analysis showed that some FtsN clusters are stationary and remained at the same position throughout the imaging time (40 s, Fig. 1f, arrow, Supplementary Movie 2). These stationary FtsN clusters likely explain the fraction of unrecovered FRAP signal. However, some FtsN clusters exhibited apparently transverse, processive movement across the short axis of the cell (Fig. 1f, arrowhead, Supplemental Movie 2). The mean directional speed measured from these kymographs was at $8.8 \pm 0.3$ nm s$^{-1}$ (Supplementary Fig. 8C, $\mu \pm$ s.e.m., $n = 92$ clusters). These directionally moving FtsN clusters likely contribute to the slow recovery rate of FRAP, as it takes ~60 s for an FtsN cluster at this speed to cross the TIRF-SIM imaging field (~500 nm, see "Methods" section). We further confirmed that the directional motion was not due to SIM imaging artifacts as we obtained the same result (Supplementary Fig. 8C, $\nu = 8.6 \pm 0.3$ nm s$^{-1}$, $\mu \pm$ s.e.m., $n = 113$ clusters) using the same mNG-FtsN fusion in conventional TIRF imaging even though the spatial resolution was lower (Supplementary Fig. 8A). The directional motion was not due to stage drifting either, because we observed both stationary and moving clusters in the same cells (Fig. 1f). Furthermore, in fixed cells, the directional, processive movement of FtsN was completely abolished (Supplementary Fig. 8B and Supplementary Fig. 8C). The combined ~9 nm s$^{-1}$ directional moving speed of FtsN clusters (Supplementary Table 6) is significantly slower than the treadmilling speed of FtsZ polymers (~30 nm s$^{-1}$)[35,36] (Fig. 1g), again demonstrating that this motion is distinct from the treadmilling dynamics of FtsZ.

## Individual FtsN molecules exhibit slow, directional motions

Apparent directional motion of a protein cluster can arise from the coordinated directional movement of individual protein molecules in the cluster or treadmilling dynamics. The latter has been reported for a few bacterial cytoskeletal proteins[58], most recently FtsZ[35–38] and PhuZ[59]. To distinguish between these two possibilities, we used 3D single-molecule tracking (3D-SMT) to investigate the movement of single FtsN molecules.

To facilitate SMT, we used a FtsN-Halo$^{SW}$ fusion (Strain EC5234 in Supplementary Table 1) that can be sparsely labeled with the bright organic dye JF646 added into the growth medium[60]. The Halo tag is inserted after amino acid E60, between the TM and E domains in the periplasm (Supplementary Fig. 1). Here we switched from N-terminal fusions to a sandwich fusion because we could use the same Halo insertion site when comparing the dynamics of full-length FtsN to those of FtsN derivatives that lack the cytoplasmic or periplasmic domain. We tracked septum-localized single FtsN-Halo$^{SW}$ molecules using a frame rate of 1 Hz and exposure time of 100 ms to effectively filter out fast, randomly diffusing molecules along the cylindrical part of the cell body. Using a custom-developed unwrapping algorithm[39,40], we decomposed 3D trajectories of individual FtsN molecules obtained from the curved cell surfaces at midcell to one-dimensional (1D) trajectories along the circumference and long axis of the cell, respectively, as previously described[40].

We found that some FtsN molecules were confined to small regions at the septum and stayed stationary (Fig. 2a, Supplementary Movie 3). Some moved directionally across the cell's short axis (Fig. 2b, Supplementary Movie 4). Some others dynamically transitioned in between different moving speeds and directions (Fig. 2c,

Supplementary Movie 5). To quantify these behaviors, we used a trajectory segmentation method[39,40] to classify segments as either stationary or moving directionally based on a statistical criterion (see "Methods" section). We found that, on average, ~55% ($55.1 \pm 1.6\%$) of the segments were classified as stationary (Fig. 2d, solid black) with an average dwell time of ~27 s ($27.3 \pm 1.3$ s, $\mu \pm$ s.e.m., $n = 315$ segments, Supplementary Table 10). For the rest of the segments, the FtsN molecules engaged in directional movement as a single population (Supplementary Fig. 9) with an average run time of ~15 s ($14.5 \pm 0.7$ s, $\mu \pm$ s.e.m., $n = 256$ segments, Supplementary Table 10) and average run speed of $9.4 \pm 0.2$ nm s$^{-1}$ ($\mu \pm$ s.e.m., Fig. 2d, solid red, Supplementary Table 10). Notably, with the two-sample Kolmogorov–Smirnov (K–S) test, the speed distribution is essentially the same as what we observed for mNG-FtsN clusters using TIRF-SIM (Supplementary Fig. 10), similar to what we recently measured for the slow-moving population of active FtsW and FtsI engaged on the sPG-track[40] (average at $9.4 \pm 0.3$ nm s$^{-1}$, Fig. 2d, red dash, Supplementary Table 6). This speed distribution has minimal overlap with FtsZ's treadmilling speed distribution under the same condition (average at $28.0 \pm 1.2$ nm s$^{-1}$, Fig. 2d, gray dash). Thus, FtsN's directional movement resembles that of the active, slow-moving population of FtsWI on the sPG-track, but not the inactive, fast-moving population of FtsWI on the FtsZ-track.

## FtsN's slow, directional movement is independent of FtsZ's treadmilling dynamics

Our previous studies have shown that the slow-moving population of FtsWI is independent of FtsZ's treadmilling dynamics but dependent on active sPG synthesis[40]. Because the speed distribution of FtsN largely superimposes with that of the slow-moving population of FtsWI (Fig. 2d), we reasoned that FtsN likely moves together with FtsWI as part of an active sPG synthesis complex. If so, we would expect that FtsN's motion depends on active sPG synthesis but not on FtsZ's treadmilling dynamics in the same manner as FtsWI.

To test whether FtsN's motion is FtsZ-dependent, we performed SMT of FtsN-Halo$^{SW}$ in four FtsZ GTPase mutant strains which show progressively slower treadmilling speeds ($ftsZ^{E238A}$, $ftsZ^{E250A}$, $ftsZ^{D269A}$, and $ftsZ^{G105S}$). We found that the average speed of directionally moving FtsN molecules in these mutants remained constant at ~9 nm s$^{-1}$ (Fig. 2e and Supplementary Table 7), independent of FtsZ's treadmilling speed (Fig. 2e, h). This behavior is essentially the same as the slow-moving, active population of FtsW and FtsI[40]. Similarly, the percentage of FtsN molecules that were moving directionally remained constant in these mutant backgrounds (Fig. 2e, h). These results demonstrate that FtsN's slow-moving dynamics are not driven by FtsZ's treadmilling dynamics.

## FtsN's slow, directional movement is independent of its cytoplasmic domain

The independence of FtsN's directional motion from FtsZ dynamics is somewhat unexpected in light of previous reports that the N-terminal cytoplasmic domain (Cyto) of FtsN can localize to the midcell through its direct interaction with the 1C domain of FtsA[17,19,20,61–66]. To address whether this or any other cytoplasmic interaction contributes to the ~9 nm s$^{-1}$ directional movement of FtsN, we constructed two FtsN mutants (Fig. 2f) with a Halo tag inserted after residue E60, the same as in the wild-type (WT) FtsN-Halo$^{SW}$. One mutant contains a D5N mutation that has been shown to reduce the interaction between FtsN and FtsA[20] (FtsN$^{D5N}$-Halo$^{SW}$, Strain EC5271 in Supplementary Table 1). In the other mutant the entire cytoplasmic and transmembrane domains were replaced with the cleavable signal sequence from DsbA for export to the periplasm (DsbA$^{ss}$-Halo-FtsN$^{\Delta Cyto-TM}$, Strain EC5263 in Supplementary Table 1). Both mutant fusions were produced in a P$_{BAD}$::$ftsN$ depletion strain grown in M9-glucose plus IPTG. Western blotting verified that native FtsN was effectively depleted and the fusions were

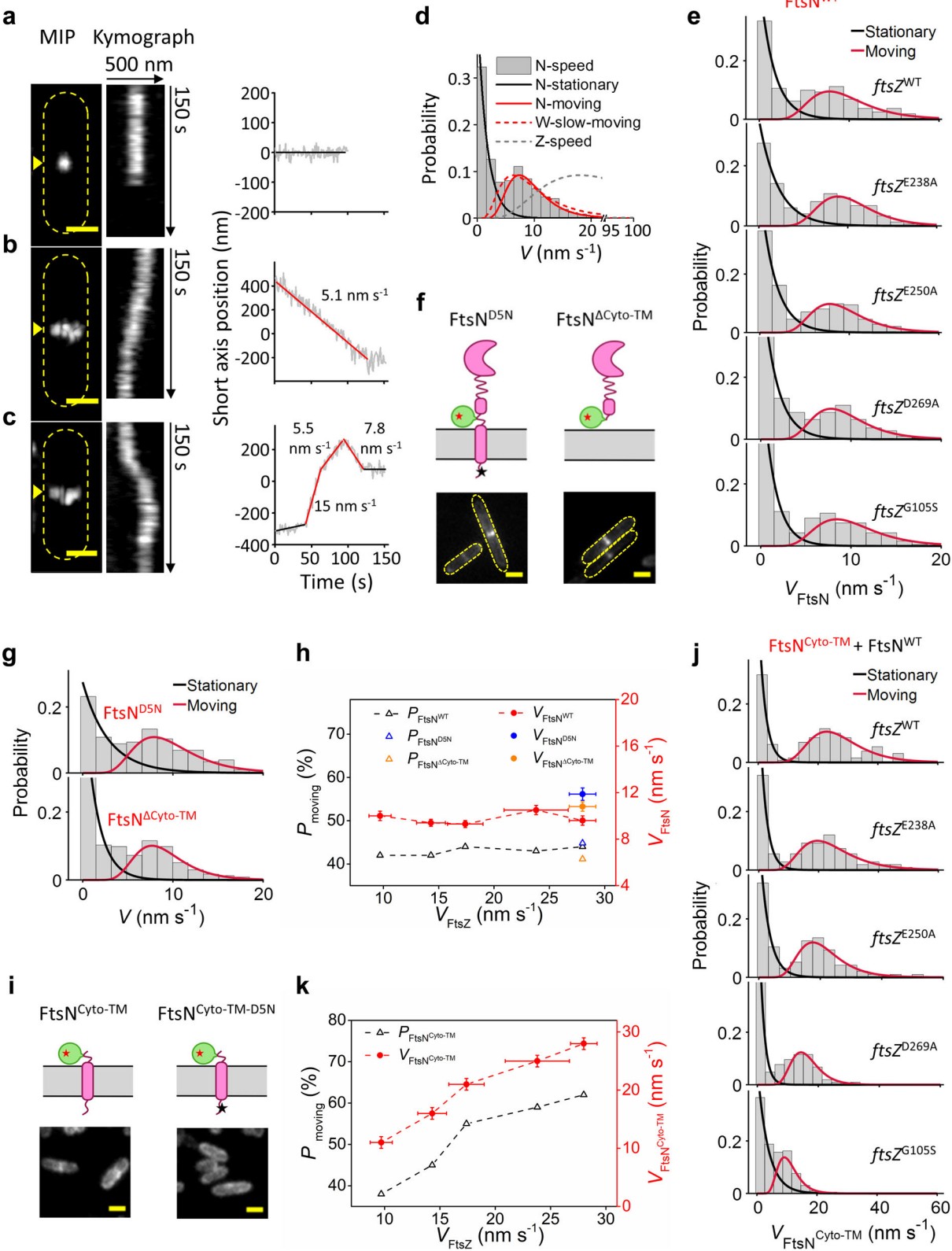

produced at physiologically appropriate levels (Supplementary Fig. 11). Both mutant fusion proteins showed prominent midcell localization and supported cell division, but cells were about twice as long as WT (Fig. 2f and Supplementary Fig. 11), likely due to delayed septum localization of FtsN and/or slowed rate of cell wall constriction. Interestingly, the percentage and speed of the directionally moving population of the mutant fusions was essentially the same as for the full-length WT FtsN-Halo^SW (Fig. 2g, h, Supplementary Table 8). These

results strongly suggest that interaction between the cytoplasmic domain of FtsN and FtsA does not contribute to the observed slow-moving dynamics of FtsN in constricting cells.

### FtsN's cytoplasmic domain exhibits fast, FtsZ treadmilling-dependent directional movement

Formally, not observing full-length FtsN on the Z-track could mean it does not localize there. But a more likely explanation is that FtsN

**Fig. 2 | FtsN exhibits a single processive moving population that is slower than, and independent of, the treadmilling dynamics of FtsZ. a–c** Representative MIPs (left), kymographs (middle) of fluorescence septal line scans (yellow arrow), and unwrapped one-dimensional positions and corresponding linear fits (right) of a stationary FtsN-Halo$^{SW}$ molecule (**a**), a directionally moving FtsN-Halo$^{SW}$ molecule (**b**), and an FtsN-Halo$^{SW}$ molecule that transitioned between different directions and speeds (**c**). Scale bars, 500 nm. Similar images were observed in $n > 100$ cells. **d** FtsN's speed distribution (gray columns) overlaid with the fit curves of the stationary (solid black) and moving (solid red) populations. Dashed curves represent the slow-moving population of FtsW molecules (red, data from a previous work[40]) and FtsZ's treadmilling speed distribution (gray, data from a previous work[36]) for comparison. The x-axis breaks from 22 to 93 nm s$^{-1}$. **e** Speed distributions of single FtsN-Halo$^{SW}$ molecules in WT and *ftsZ* GTPase mutant strains overlaid with corresponding fit curves. **f** Schematic representation of FtsN$^{D5N}$-Halo$^{SW}$ (left) and Halo-FtsN$^{ΔCyto-TM}$ (right) and representative fluorescence cell images (bottom). Green

bubble; Halo tag; red star: JF646 dye. Scale bars, 1 μm. Similar images were observed in $n > 40$ cells for each mutant. **g** Speed distributions of single FtsN$^{D5N}$-Halo$^{SW}$ (top) and Halo-FtsN$^{ΔCyto-TM}$ (bottom) molecules overlaid with corresponding fit curves. **h** Percentage of moving population (black triangle) and average moving speed (red circle) of FtsN are independent of FtsZ's treadmilling speed. FtsN$^{D5N}$ and FtsN$^{ΔCyto-TM}$ data are shown in blue and orange, respectively. **i** Schematic representation of FtsN$^{Cyto-TM}$-Halo$^{SW}$ (left) and FtsN$^{Cyto-TM-D5N}$-Halo$^{SW}$ (right) with representative fluorescence cell images (bottom). Scale bars, 1 μm. Similar images were observed in $n > 40$ cells for each mutant. **j** Speed distributions of single FtsN$^{Cyto-TM}$-Halo$^{SW}$ molecules in WT and *ftsZ* GTPase mutant strains overlaid with corresponding fit curves. **k** Percentage of moving population (black triangle) and average moving speed (red circle) of FtsN$^{Cyto-TM}$ are dependent on FtsZ's treadmilling speed. All data are presented as mean ± s.e.m. The sample sizes of all data points in (**h**) and (**k**) are listed in Supplementary Tables 7, 8 and 9. Source data are provided as a Source Data file.

---

localizes to the Z-track only transiently and in low abundance before the FtsN·FtsA interaction in the cytoplasm during early divisome assembly is supplanted during constriction by (presumably stronger) interactions of FtsN's periplasmic domain with other division proteins and/or sPG. This hypothesis predicts that preventing FtsN's periplasmic interactions should give rise to accumulation a detectable population of fast, directionally moving FtsN molecules that are end-tracking on treadmilling FtsZ polymers via FtsN's interaction with FtsA.

To examine this possibility, we constructed a FtsN$^{Cyto-TM}$-Halo$^{SW}$ fusion, in which the E and SPOR domains of FtsN are removed and a Halo tag is inserted at the same position (E60) as the full-length sandwich fusion (Fig. 2i, Strain EC5317 in Supplementary Table 1). Because FtsN$^{Cyto-TM}$ cannot support cell division by itself, we expressed it ectopically from the chromosome in the presence of WT FtsN. Ensemble fluorescence imaging showed that FtsN$^{Cyto-TM}$-Halo$^{SW}$ exhibits patchy fluorescence along the cell perimeter and has markedly decreased midcell localization compared to full-length FtsN (Fig. 2i). This observation is consistent with FtsN$^{Cyto-TM}$ having a transmembrane domain but not the SPOR domain, which is the major septum localization determinant[11,45]. Further mutating the conserved D5 residue in the cytoplasmic domain (FtsN$^{Cyto-TM-D5N}$-Halo$^{SW}$, Strain EC5321 in Supplementary Table 1) completely abolished any residual midcell localization (Fig. 2i), demonstrating that the limited midcell localization of FtsN$^{Cyto-TM}$-Halo$^{SW}$ is indeed mediated by FtsN's interaction with FtsA and not by interaction (e.g., potential dimer/oligomer formation) with WT FtsN molecules in the cells.

Despite the poor septal localization, we were able to track remaining single FtsN$^{Cyto-TM}$-Halo$^{SW}$ molecules at the midcell in a series of FtsZ GTPase WT and mutant backgrounds. Strikingly, we found that now in ~60% (62.5 ± 1.9%) of the SMT segments FtsN$^{Cyto-TM}$-Halo$^{SW}$ molecules moved at an average speed of ~30 nm s$^{-1}$ (29.1 ± 1.7 nm s$^{-1}$, μ ± s.e.m., $n = 130$ segments, Fig. 2j and Supplementary Table 9) in the FtsZ WT background, similar to FtsZ's treadmilling speed. In four FtsZ GTPase mutant strains (*ftsZ*$^{E238A}$, *ftsZ*$^{E250A}$, *ftsZ*$^{D269A}$, and *ftsZ*$^{G105S}$), we observed progressively reduced speed and population percentage of directionally moving FtsN$^{Cyto-TM}$-Halo$^{SW}$ (Fig. 2k and Supplementary Table 9). There was no discernible slow-moving population of FtsN$^{Cyto-TM}$-Halo$^{SW}$ under any of these conditions. There was, however, a significant stationary population, which could be explained by the association of FtsN$^{Cyto-TM}$-Halo$^{SW}$ with FtsA bound to internal sites in FtsZ filaments, as we documented previously for a fraction of septal FtsW molecules[40]. Consistent with this possibility, the average lifetime of these stationary FtsN$^{Ctyo-TM}$-Halo$^{SW}$ molecules is in the range of 12–19 s (Supplementary Table 9), significantly shorter than that of the full-length FtsN-Halo$^{SW}$ fusion (~30 s), but similar to that of stationary FtsW molecules and FtsZ subunits under the same growth and imaging conditions[40].

Taken together, these results strongly suggest the FtsN−FtsA interaction drives the FtsZ treadmilling-dependent end-tracking

behavior of FtsN in pre-divisional cells, but this interaction (and thus end-tracking) is diminished once FtsN's periplasmic interactions take over with the onset of constriction. In other words, not observing full-length FtsN-Halo$^{SW}$ on the Z-track probably reflects the technical difficulties of observing a low-abundance population of FtsN molecules present in only a subset of cells. A previous in vitro study showed that the membrane-anchored cytoplasmic domain of FtsN is capable of following treadmilling FtsZ polymers through a diffusion-and-capture mechanism[67], but does not directionally end-track FtsZ at the single-molecule level as what we observed here. This difference is most likely due to the more restricted diffusion of FtsN$^{Cyto-TM}$ along the septum area in vivo compared to in vitro, as we previously predicted in the Brownian ratchet model[39].

## FtsN's directional movement depends on sPG synthesis

Our results so far demonstrated that the slow, directional movement of full-length FtsN is independent of FtsZ's treadmilling dynamics. To examine whether it is driven by active sPG synthesis like the slow-moving population of FtsWI[40], we performed SMT of FtsN-Halo$^{SW}$ under conditions of altered sPG synthesis activity.

We first examined the effect of inhibiting FtsW's glycosyltransferase (GTase) activity on the movement of FtsN-Halo$^{SW}$ using a functional FtsW variant, FtsW$^{I302C}$, which can be specifically inhibited upon the addition of the cysteine-reactive reagent MTSES (2-sulfonatoethylmethanethiosulfonate)[40]. In this strain background, FtsN-Halo$^{SW}$ exhibited similar dynamics as in the parent FtsW$^{WT}$ cells (Fig. 3a, top two panels, Supplementary Table 10). In the presence of MTSES (100 μM, 60 min), however, the directionally moving population of FtsN-Halo$^{SW}$ was significantly reduced and on average ~80% of segments were stationary (80.4 ± 1.4%, $n = 115$ segments, Fig. 3a, c, Supplementary Table 10). Note that the addition of MTSES in the parent FtsW$^{WT}$ background did not produce any appreciable change in FtsN-Halo$^{SW}$ dynamics (Supplementary Fig. 12, Supplementary Table 10). Strikingly, MTSES depleted the slow-moving population of FtsN-Halo$^{SW}$ in the FtsW$^{I302C}$ background to essentially the same extent as it depleted the slow-moving population of FtsW$^{I302C}$ itself in a previous report[40]. We conclude that the directional motion of FtsN is coupled to FtsW's GTase activity.

Next, we tracked the movement of FtsN-Halo$^{SW}$ in the presence of aztreonam, an antibiotic that specifically inhibits the transpeptidase (TPase) activity of FtsI[68]. In cells treated with aztreonam (1 μg ml$^{-1}$, 30 min), the directionally moving population of FtsN was again substantially reduced and ~90% of FtsN's SMT segments were stationary at the septum (Fig. 3b, c, Supplementary Table 10). In addition, depleting the cell wall precursor Lipid II using Fosfomycin (inhibits the essential lipid II synthesis enzyme MurA[69], 200 μg ml$^{-1}$, 30 min) resulted in near complete abolishment of the directionally moving population of FtsN, approaching the background level in fixed cells (Fig. 3b, c, Supplementary Table 10). All these behaviors are, again, identical to

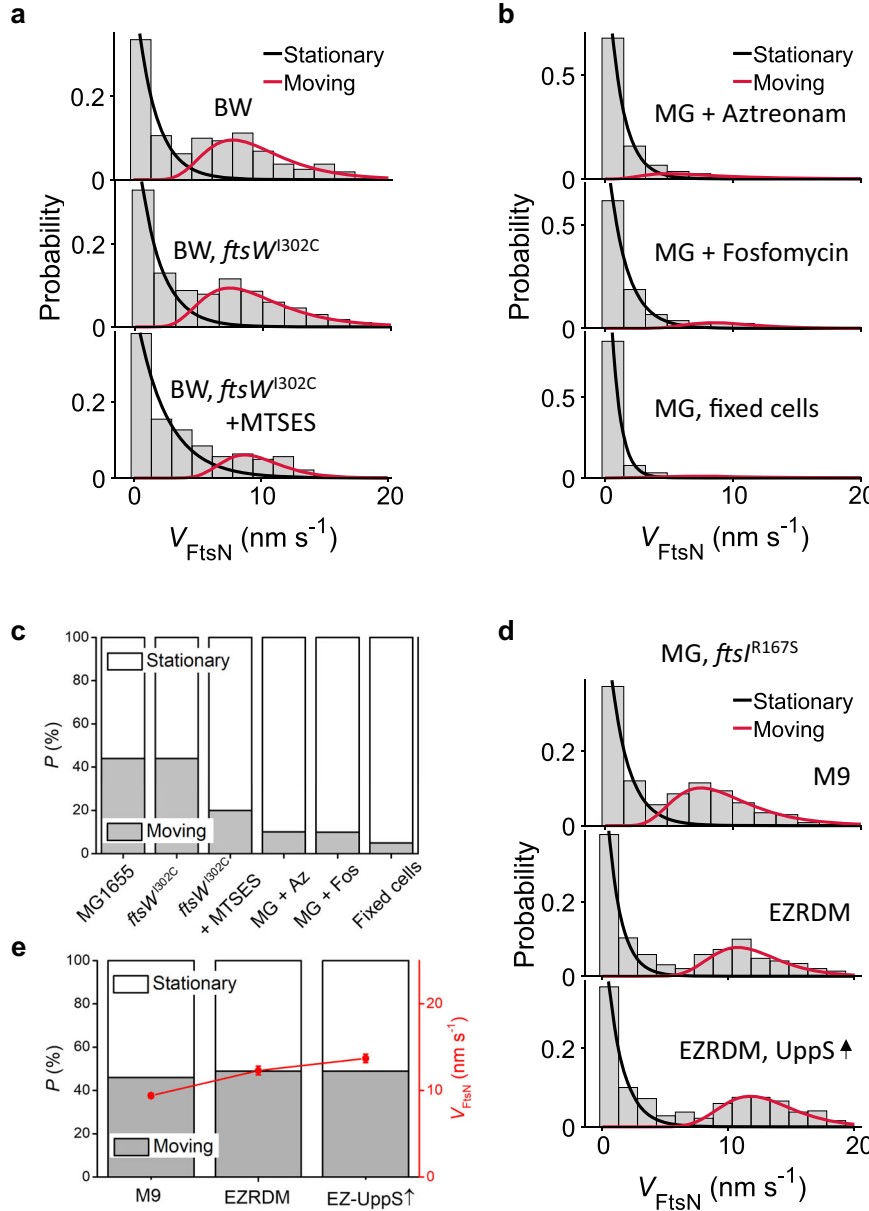

**Fig. 3 | FtsN's processive moving population is driven by sPG synthesis activity.**
a Speed distributions and the corresponding fit curves of the stationary (black)
and moving (red) populations of single FtsN-Halo[SW] molecules in the BW25113
WT (top) and *ftsW*[I302C] variant strain in the absence (middle) or presence
(bottom) of MTSES. **b** Speed distributions of single FtsN-Halo[SW] molecules in
the MG1655 WT strain treated with aztreonam (top) or fosfomycin (middle).
Fixed cells without antibiotic treatment are shown as a control (bottom).
**c** Percentage of the processively moving population of FtsN (gray bar) gra-
dually decreased when sPG synthesis is inhibited under the conditions in

(**a**) and (**b**). **d** Speed distributions of single FtsN-Halo[SW] molecules in the
MG1655 *ftsI*[R167S] superfission variant strain background grown in M9-glucose,
EZRDM or in EZRDM medium with UppS overproduction (top to bottom).
**e** Percentage of the processive moving population (gray bar) and average
moving speed (red circle) under conditions in (**d**). The data of average speed
are presented as mean ± error, where the error is the standard deviation from
200 bootstrap samples pooled from three independent experiments. The
sample size of each point is listed in Supplementary Tables 10. Source data are
provided as a Source Data file.

the depletion of the slow-moving population of FtsW under identical
conditions as we previously observed[40].

To probe the dynamics of FtsN under conditions of enhanced cell
wall synthesis, we made use of an *ftsI*[R167S] superfission strain, which
partially alleviates the need for FtsN[40]. Previously we showed that by
growing *ftsI*[R167S] cells in a rich defined medium (EZRDM) and over-
expressing the undecaprenyl pyrophosphate synthetase (UppS, an
enzyme responsible for making Lipid II[70]), the percentage of direc-
tionally moving FtsW molecules on the slow sPG-track increased to
nearly 100% and their speed increased to ~13 nm s[-1], likely reflecting the
in vivo sPG elongation rate[40]. If FtsN is in complex with FtsWI and its
movement is coupled to FtsWI's activity, we should observe similar

changes in FtsN's dynamics. Indeed, the average speed of FtsN-Halo[SW]
accelerated from 9.4 ± 0.3 nm s[-1] in M9 to 12.3 ± 0.5 nm s[-1] in EZRDM
and further to 13.7 ± 0.5 nm s[-1] in EZDRM with concomitant over-
production of UppS (Fig. 3d, e, Supplementary Table 10). These
increased speeds are similar to those of the slow-moving population of
FtsW under the same conditions[40]. Most importantly, the distributions
of the speed, processive run length and run time of FtsN-Halo[SW] are
indistinguishable from those of FtsW under the EZRDM and UppS
overexpression conditions, where FtsW essentially only exhibits one
slow-moving population (Supplementary Fig. 13 and Supplementary
Table 11). These results strongly suggest that FtsN forms a processive
sPG synthesis complex with active FtsWI.

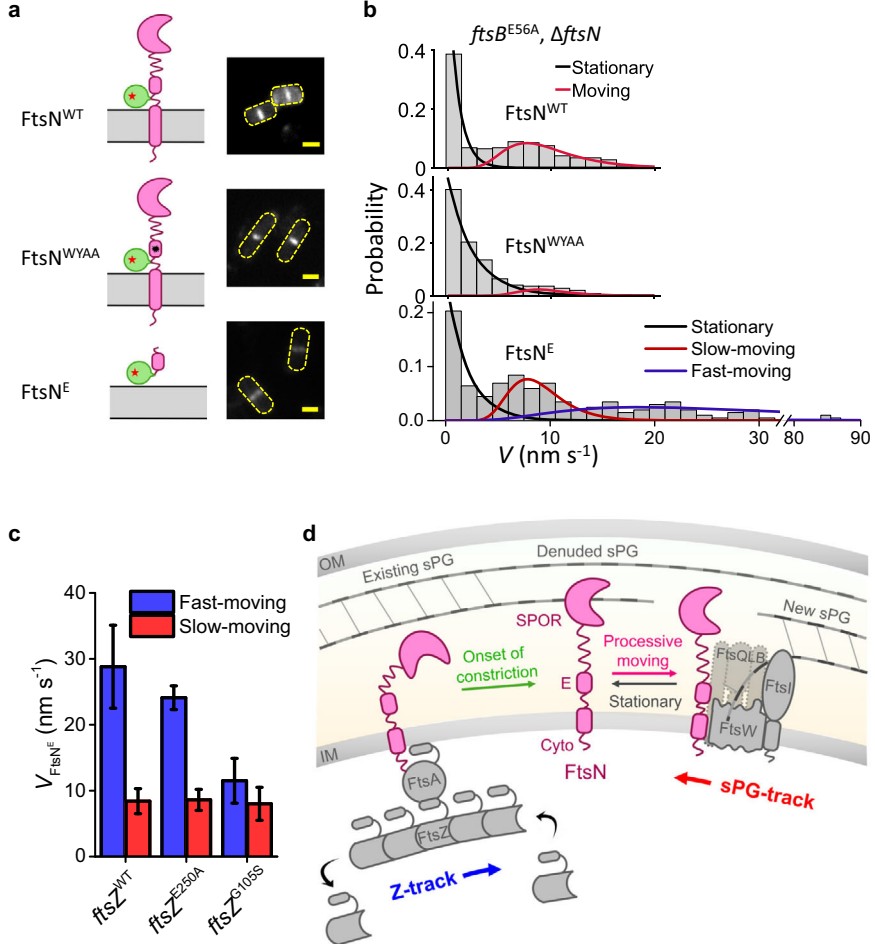

**Fig. 4 | FtsN's E domain is sufficient to form a processive complex with FtsWI on the sPG-track. a** Schematic representation of FtsN-Halo[SW], FtsN[WYAA]-Halo[SW], and Halo-FtsN[E] fusions (left, black star in FtsN[WYAA]-Halo[SW] represents the W83A and Y85A double substitution) and the corresponding representative ensemble fluorescence images (right, all expressed in the superfission variant *ftsB*[E56A]Δ*ftsN* background). Scale bars, 1 μm. Similar images were observed in *n* > 50 cells for each mutant. **b** Speed distributions and corresponding fit curves for stationary (black), slow-moving (red) and fast-moving (blue) populations of single FtsN-Halo[SW], FtsN[WYAA]-Halo[SW], and Halo-FtsN[E] molecules (top to bottom). The *x*-axis breaks from 32 to 79 nm s[-1]. **c** Decomposed mean speed of the fast-moving population of Halo-FtsN[E] (blue bars) is correlated with FtsZ's GTPase activity, while the speed of the slow-moving population of Halo-FtsN[E] (red bars) is independent of FtsZ's GTPase activity. Data are presented as mean ± s.e.m. and listed in Supplementary Table 12. Source data are provided as a Source Data file. **d** A model depicting how FtsN

activates sPG synthesis. At the onset of constriction, before denuded glycans have accumulated in sPG, FtsN is recruited through the interaction of its cytoplasmic tail with FtsA, and is distributed around the septum by treadmilling FtsZ polymers. After the onset of constriction, FtsN is recruited primarily by binding of its SPOR domain to denuded glycan strands. The SPOR-glycan interaction out competes the interaction between FtsN and FtsA and creates a pool of stationary FtsN at the septum. Release of FtsN from denuded glycans and interaction of FtsN's E domain with FtsWI either directly or through FtsQLB (grayed out) results in the formation of an activated sPG synthesis complex, which engages in processive sPG synthesis. The active complex is sustained on the sPG-track by the presence of FtsN in the complex. Stochastic or regulated dissociation of FtsN from the complex results in the termination of sPG synthesis. FtsWI can be released to the fast Z-track, but FtsN preferentially rebinds with denuded glycan strands, waiting for the next activation event.

## FtsN's E domain mediates the formation of a processive complex with FtsWI on the sPG-track

What interaction mediates the processive complex between FtsN and FtsWI? Past studies have shown that a short fragment of FtsN comprising only the second helix in the periplasmic E domain is both necessary and sufficient for cell division when overexpressed[11,19]. An FtsN mutant containing changes in two conserved amino acids in the E domain (WYAA, with W83 and Y85 changed to alanines) fails to support cell division, presumably because these residues are important for binding of FtsN to the FtsWI complex[19,71]. If so, the WYAA mutant protein might not be able to form a processive complex with FtsWI, resulting in cell division failure. Because the WYAA mutant is lethal due to the lack of FtsWI activity, to test this hypothesis, we took advantage of an *ftsB* superfission strain (*ftsB*[E56A] Δ*ftsN*) in which FtsWI is constitutively active without FtsN[19].

We first constructed a FtsN[WYAA]-Halo[SW] fusion and expressed it from a plasmid in the superfission variant *ftsB*[E56A] Δ*ftsN* background (Strain JL398 in Supplementary Table 1). As a control, we also expressed the WT FtsN-Halo[SW] in the same strain background (Strain JL397 in Supplementary Table 1). We observed that both FtsN[WYAA]-Halo[SW] and WT FtsN-Halo[SW] exhibited similar levels of midcell localization (Fig. 4a), as FtsN's major localization determinant—the SPOR domain—remains intact in both fusion proteins. However, the majority of FtsN[WYAA]-Halo[SW] fusion protein remained stationary at septa as the directionally moving population was significantly diminished to ~11% compared to that of WT FtsN-Halo[SW] (~44%) (Fig. 4b, Supplementary Table 12). Combined with our previous observation that FtsW's slow-moving population is also significantly reduced in this strain background even though FtsN is no longer essential[40], this finding suggests that the formation of the processive sPG synthesis complex between FtsN and

FtsWI is indeed mediated by the two conserved residues and crucial for activating FtsWI.

Finally, to address directly whether the E domain itself is sufficient for the processive movement of FtsN, we tracked the dynamics of a Halo fusion to only the E domain (amino acids 61 to 105, containing helix 1 and the essential helix 2) in the same $ftsB^{E56A}$ $\Delta ftsN$ strain background (Strain JL399 in Supplementary Table 1). Despite the absence of a SPOR domain, Halo-FtsN$^E$ exhibited convincing, albeit weak, midcell localization (Fig. 4a), demonstrating that interaction of the E domain with the sPG synthesis complex is sufficient for detectable septal localization. Most interestingly, septal Halo-FtsN$^E$ moved processively in ~63% of the SMT segments (Fig. 4b), and a new, fast-moving population (~70% of all moving segments, $v = 28.8 \pm 6.3$ nm s$^{-1}$, $\mu \pm$ s.e.m., $n = 127$ segments, Supplementary Table 12) emerged in addition to the slow-moving population (29.8 ± 13.7%, $v = 8.4 \pm 1.9$ nm s$^{-1}$, $\mu \pm$ s.e.m., $n = 75$ segments, Supplementary Table 12). These two populations closely resemble the fast- and slow-moving populations of FtsW and FtsI on the Z- and sPG-tracks, respectively (Supplementary Fig. 14). Importantly, we further confirmed that the fast-moving population of Halo-FtsN$^E$ is indeed on the Z-track by showing that its velocity was reduced in two FtsZ GTPase mutants with diminished treadmilling dynamics ($ftsZ^{E250A}$, $ftsZ^{G105S}$, Fig. 4c, Supplementary Fig. 14B, Supplementary Table 12). Both the percentage and mean dwell time of stationary FtsN$^E$ molecules increased with reduced FtsZ GTPase activity (Supplementary Fig. 15, Supplementary Table 12), in line with the increased FtsZ filament length in FtsZ GTPase mutants, and similar to those of the slow-moving population of FtsW under identical conditions[40]. These results demonstrate that the E domain is sufficient to form a processive complex with FtsWI, and such a complex can be maintained even on the Z-track when the SPOR domain is absent. In other words, the SPOR domain may be the major determinant to prevent the release of the sPG synthesis complex from the sPG-track to the Z-track.

## Discussion

FtsN, a late recruit to the *E. coli* divisome, works through FtsA and the FtsQLB complex to activate synthesis of septal PG by FtsWI. Previous work has shown that FtsWI moves directionally around the circumference of the division site on two tracks, one driven by FtsZ treadmilling (Z-track), the other driven by sPG synthesis (sPG-track). Only FtsWI on the sPG-track is actively engaged in sPG synthesis. Previous work also revealed that FtsN activates FtsWI by redistributing it from the Z-track to the sPG-track, but how FtsN does so is unclear. Here we show that (1) FtsN is an essential component of the processive FtsWI complex on the sPG-track; (2) FtsN's essential (E) domain is both necessary and sufficient to maintain active FtsWI complexes on the sPG-track; and (3) FtsN's PG-binding SPOR domain prevents FtsWI from transitioning back to the fast-moving Z-track, hence increasing the fraction of FtsWI complexes engaged in sPG synthesis.

We observed that FtsN forms a discontinuous or patchy ring-like structure and exhibits distinct septal organization and dynamics compared to those of the FtsZ-ring. FtsN-rings were first visible as such at a septal diameter of ~600 nm while FtsZ-rings appeared at ~950 nm. The difference in their timing of ring assembly could reflect the fact that the small amounts of FtsN recruited at the onset of constriction do not create the appearance of a ring, which only occurs after sufficient denuded glycans accumulate in the nascent division septum to recruit a larger amount of FtsN. Thus, the 600 nm diameter may reflect the transition from primarily FtsA-mediated to primarily denuded glycan-mediated FtsN localization[17]. We further determined that FtsN is present at ~300 molecules per cell under our growth conditions and found that even at maximal septal accumulation <20% of these molecules localize to the FtsN-ring. These numbers imply the FtsN-ring contains at most ~60 FtsN molecules. It will be interesting to learn whether this ratio holds true for FtsWI and other late divisome proteins, as these values place constraints on the number of active sPG synthesis complexes and their stoichiometries.

About half of FtsN molecules in the FtsN-ring are essentially static, with an average dwell time of ~27 s. The stationary population of FtsN was dramatically reduced in constructs lacking the SPOR domain, indicating stationary FtsN is anchored to denuded glycans in sPG. Further studies will be needed to determine whether the ~27 s dwell time reflects the off-rate for the releasing of the SPOR domain from denuded glycans or their turnover rate by lytic transglycosylases. A potential point of confusion is that we also observed a small population of stationary FtsN molecules during SMT of truncated fusions that lack the SPOR domain (SPOR-truncation) and thus cannot bind sPG. As indicated by their dwell time, these stationary FtsN SPOR-truncations are likely bound to FtsA and/or FtsWI complexes associated with internal positions in FtsZ polymers, as we have demonstrated previously for FtsWI[40]. Western blotting demonstrated that cells expressing FtsN SPOR-truncations were nearly devoid of full-length FtsN under our growth conditions (Supplementary Fig. 11), arguing against the possibility that stationary FtsN SPOR-truncation molecules are bound to residual full-length FtsN.

The other half of septal FtsN molecules move processively at a speed of ~9 nm s$^{-1}$. Numerous lines of evidence indicate these FtsN molecules are bound to FtsWI complexes engaged in synthesis of sPG on the sPG-track. The 9 nm s$^{-1}$ velocity is essentially identical to that of the slow-moving population of active FtsWI and much slower than the ~30 nm s$^{-1}$ velocity of treadmilling FtsZ. What's more, as shown previously for FtsWI on the sPG-track, the speed and fraction of FtsN molecules moving processively were increased under conditions that increase the rate of sPG synthesis and decreased by impeding sPG synthesis, but insensitive to perturbations of the treadmilling speed of FtsZ. Not only the average speed of FtsN, but also its speed distribution, average run time and average run length, are similar to those of FtsWI.

Tracking of various mutant derivatives of FtsN revealed that the only domain required for processive movement on the sPG-track is the essential (E) domain, which is proposed to interact with FtsWI, likely via the FtsQLB complex[15,16,18,19]. Most importantly, such a complex is crucial for activating and sustaining sPG synthesis in a processive manner, as a double point mutation that inactivates the E domain (WYAA) prevents formation of the processive complex and causes failure of cell division. These findings imply FtsN must persist as part of these complexes to maintain their activity on sPG-track. Although the E domain has also been implicated in binding to the bifunctional PG synthases PBP1a and PBP1b[72–74], these enzymes are not known to move processively[13,75], so they are not strong candidates to account for the directional movement of FtsN. They could, however, interact with the stationary population of septal FtsN, which requires further investigation.

Somewhat unexpectedly, we did not observe a population of full-length FtsN molecules moving at ~30 nm s$^{-1}$ on the Z-track, suggesting the FtsN-FtsA interaction is not a major FtsWI activation pathway once constriction has commenced. In support of this notion, we found that abrogating the FtsN-FtsA interaction, either by deleting the cytoplasmic domain or introducing a D5N substitution, did not diminish the slow-moving population of FtsN molecules on the sPG-track (Fig. 2f–h). Nevertheless, cells were two- to four-fold longer than wild-type, likely due to delayed initiation of constriction. Conversely, FtsN SPOR-truncation constructs (FtsN$^{Cyto-TM}$ and FtsN$^E$) localized to the Z-track even in constricting cells. These constructs likely mimic the behavior of full-length FtsN at early stages of division before amidase processing of sPG has created denuded glycans for SPOR domains to bind to. Collectively, our FtsN tracking data add to previous evidence that the FtsN-FtsA interaction is important early in the division process and imply that the interaction between FtsN's SPOR domain and denuded glycans is stronger than that between FtsN's cytoplasmic domain and FtsA. Finally, the SPOR domain serves not only to increase

the local concentration of FtsN at the septum, but also has a previously unrecognized role in preventing FtsWI complexes from transitioning to the Z-track, where they become inactive. How the SPOR domain would prevent FtsWI from moving to the Z-track is not immediately obvious and requires further investigations.

We have updated the currently accepted model for FtsN function to incorporate the findings presented here (Fig. 4d). FtsN is first recruited to the septum through the interaction between its cytoplasmic tail with FtsA, and is distributed around the septum by treadmilling FtsZ polymers. After the onset of constriction, FtsN binds to denuded glycan strands through its SPOR domain, which diminishes the interaction between FtsN and FtsA and creates a pool of stationary FtsN molecules at the septum. The interaction between FtsN's E domain with FtsWI (either directly or through FtsQLB) mediates the formation of an activated sPG synthesis complex that engages in processive sPG synthesis. Presumably, FtsN has to release its hold on denuded glycans to move processively with FtsWI; such release might happen spontaneously or be triggered by interaction of the E domain with FtsWI. Subsequently, stochastic or regulated dissociation of FtsN from the synthesis complex results in the termination of sPG synthesis, which could release FtsWI back to the fast Z-track. Dissociated FtsN could immediately rebind with denuded glycan strands, waiting for the next activation event. According to this model, the major function of the SPOR domain is to maintain FtsWI on the sPG-track, which it does by preventing FtsN and FtsWI from diffusing away from the septum or reassociating with the fast-moving FtsZ-track. These new possibilities about FtsN's function will be the subject of future studies.

## Methods
### Growth media
Lysogeny broth (LB) was employed for routine growth and genetic manipulation of *E. coli* strains. For microscopy, cells were grown in the rich defined medium EZRDM[76] or M9-glucose minimal medium (0.4% D-glucose, 1× MEM amino acids and 1× MEM vitamins)[77]. For single-molecule tracking experiments, vitamins were omitted from the M9 medium (termed M9$^-$ minimal medium) to minimize background. Where appropriate, antibiotics were included as follows: ampicillin, $100\,\mu g\,ml^{-1}$; carbenicillin, $25\,\mu g\,ml^{-1}$; chloramphenicol, $35\,\mu g\,ml^{-1}$; kanamycin, $40\,\mu g\,ml^{-1}$; spectinomycin, $100\,\mu g\,ml^{-1}$ for plasmids and $35\,\mu g\,ml^{-1}$ for chromosomal alleles. In general, antibiotics were used when propagating *E. coli* strains in LB but omitted when growing cells in minimal media for microscopy. This was possible because most of the fusions used in this study were integrated into the chromosome. However, chloramphenicol was added to minimal media for growth of strains containing the plasmids pJL132, pJL133, and pJL136. Where appropriate, 0.2% L-arabinose was used to express *ftsN* under P$_{BAD}$ control[78]. Isopropyl β-D-1-thiogalactopyranoside (IPTG) was used as indicated to express genes under control of modified (weakened) *Trc* promoters.

### Bacterial strains, oligonucleotide primers, and plasmids
Standard procedures were used for analysis of DNA, PCR, electroporation, transformation, P1 transduction and integration of CRIM plasmids[79]. Bacterial strains are listed in Supplementary Table 1, which also describes strain construction. Plasmids are described in Supplementary Table 2, followed by descriptions of how these plasmids were made. Some plasmids were assembled by amplifying appropriate DNA fragments using Q5 DNA polymerase followed by ligation or assembly into restriction-digested vectors using T4 DNA ligase or NEBuilder HiFi DNA Assembly Master Mix, respectively (New England Biolabs). Alternatively, plasmids were assembled by amplifying appropriate insert and vector DNA fragments followed by In-Fusion cloning (Takara, In-Fusion HD Cloning Kit). The Quikchange Lightening Kit (New England Biolabs) was used for site-directed mutagenesis as needed. Oligonucleotides were from Integrated DNA Technologies

(Coralville, IA), and are listed in Supplementary Table 3. Regions of plasmids encompassing fusion genes constructed by PCR were verified by DNA sequencing.

### Purification of His$_6$-FtsN periplasmic domain
A fusion of a hexahistidine tag to the periplasmic domain of FtsN (residues 49–319) was overproduced in BL21(DE3) and purified on Talon affinity resin according to instructions from the manufacturer (Takara Bio USA, Inc.). Purified protein was dialyzed into 50 mM Na$_2$HPO$_4$, 200 mM NaCl, 5% glycerol, pH 7.5. Protein concentration was determined by BCA assay (Pierce) with BSA as standard. The yield from 1 Liter of cells was 7 mg at a concentration of 3.5 mg/ml and 80% purity as estimated from SDS-PAGE.

### Anti-FtsN anti-sera
Rabbit anti-FtsN was raised against a maltose-binding protein fusion to the periplasmic domain of FtsN (residues 56-319) and has been described[80]. Because FtsN co-migrates in SDS-PAGE with maltose-binding protein, it was necessary to remove anti-MBP antibodies. This was accomplished by incubating the anti-serum with a concentrated cell lysate from the *E. coli* MBP overproduction strain DH5α/pMAL-c2 as described[81].

### Growth of cells for various experiments
Cultures were grown differently depending on the experiment for which the cells were to be used.

Our standard procedure to grow cells for superresolution or SMT microscopy was as follows. Starter cultures were grown overnight at 30 °C or 37 °C in LB, supplemented in most cases with 0.2% L-arabinose (to induce P$_{BAD}$::*ftsN*) and antibiotics if appropriate. The next day cells were washed once with M9-glucose to remove antibiotics and arabinose, then diluted 500 to 2000-fold into 3 ml M9-glucose containing IPTG as indicated to induce expression of chromosomal *ftsN* fusions; antibiotics were omitted except for plasmid strains. Cultures were incubated at room temperature with shaking until the OD$_{600}$ reached ~0.35 (~18 h).

For the complementation assay in Supplementary Fig. 2A and Supplementary Fig. 3A, overnight cultures were diluted into LB (no arabinose) to OD$_{600}$ = 0.1, incubated for ~30 min until they reached OD$_{600}$ = 0.2, then 10-fold serial dilutions were prepared in M9-glucose. Dilutions were spotted onto M9-glucose plates, which were photographed after 18 hr incubation at 37 °C. To obtain the growth curves in Supplementary Fig. 2B, overnight cultures were grown in M9-glucose. The following day, OD$_{600}$ was measured with a Nanodrop and cultures were diluted to an initial OD$_{600}$ of 0.1 in 200 μl M9-glucose in a Corning Costar sterile 96-well plate. The 96-well plates were incubated and shaken in a Tecan Infinite M200 Pro set at 30 °C, with OD$_{600}$ measurements taken every 30 min for 23.5 h, shaking the plate for 3 min at 220 rpm before measuring. Doubling times were calculated from the linear phase of the log-transformed growth curve data as fitted with a straight line.

For the localization experiment in Supplementary Fig. 2C, starter cultures were grown overnight at 30 °C in LB supplemented with antibiotics and 0.2% L-arabinose to induce P$_{BAD}$::*ftsN*. The next day cells were washed once with M9-glucose to remove arabinose, then diluted 500 to 2000-fold into 3 ml M9-glucose containing antibiotics to select for plasmids but without IPTG (i.e., no induction was need because leaky expression from the plasmid is sufficient). Cultures were incubated at room temperature with shaking until the OD$_{600}$ reached ~0.35 (~18 h).

For the Western blots in Supplementary Fig. 4, cultures were grown similarly to obtain samples except that antibiotics were omitted for chromosomal fusions (Supplementary Fig. 4A–C) and IPTG was included as indicated in the figure. At the time cells were harvested for microscopy, a 0.5 ml aliquot of each culture was fixed with paraformaldehyde for cell length determinations. Cells were photographed

under phase contrast and measured using Olympus cellSens Dimension software.

## Western blotting

Cells from 1 ml of culture at an $OD_{600}$ ~ 0.35 were harvested by centrifugation and the cell pellet was taken up in ~70 μl 1x Laemmli Sample Buffer (LSB) to achieve a sample concentration of 5.0 $OD_{600}$ units per ml. Samples were heated for 10 min at 95 °C before loading 10 μl onto a precast mini-PROTEAN TGX gel (10% polyacrylamide, from Bio-Rad, Hercules, CA). Electrophoresis, transfer to nitrocellulose and blot development followed standard procedures. Primary antibody was a 1:1000 dilution of polyclonal rabbit anti-FtsN sera that had been pre-absorbed against a lysate of DH5α/pMAL-C2 as described above. Secondary antibody was horseradish peroxidase-conjugated goat anti-rabbit antibody (1:10,000; Pierce, Rockford, IL), which in turn was detected with SuperSignal WEST Pico Plus chemiluminescent substrate (Pierce, Rockford, IL). Blots were visualized with a ChemDoc Touch Imaging System (BioRad, Hercules, CA).

## Quantification of FtsN

The wild-type strains EC251 and BW25113 were grown at room temperature in M9-glucose to $OD_{600}$ ~0.35 as described above. Multiple 1 ml aliquots were harvested by pelleting cells in a microfuge and taking up pellet in LSB to achieve a sample concentration of 5.0 $OD_{600}$ units per ml. Samples were pooled. In parallel, cultures were diluted and plated to determine CFUs.

To create a standard curve, purified $His_6$-FtsN periplasmic domain was mixed with an aliquot of cell extract and then serially diluted into the extract. After heating, 10 μl samples (corresponding to $1.9 \times 10^7$ cells of MG1655 or $1.7 \times 10^7$ cells of BW25113) were loaded onto 10% polyacrylamide gels. Subsequent steps followed the Western blotting procedures described above. Band intensities were quantified using ImageJ and used to interpolate ng of native FtsN on the blot, which was converted to number of molecules using the molecular masses of $His_6$-$FtsN^{peri}$ (31,869 kDa) and native FtsN (35,793 kDa).

EC251 was determined to have on average 0.27 ng per lane. EC251 contained on average 264 molecules per cell ($N = 2$). BW25113 was determined to have on average 0.32 ng per lane. BW25113 contained on average 310 molecules per cell ($N = 2$).

## Construction of functional FtsN fusions

FtsN has at least four functional domains spanning from the N-terminal cytoplasmic tail to the C-terminal periplasmic SPOR domain. To avoid any potential interference from the tag, we screened 11 FtsN fusions that had mNeonGreen (mNG) fused to the N-terminus, C-terminus or inserted at internal positions of FtsN. The complementation, cell growth rates, and midcell localization images during cell division were obtained as the criteria to identify the functional fusions. Finally, the fusions with mNG fused to the N-terminus (mNG-FtsN) or inserted between E60 and E61 (E60-mNG-E61) passed all the tests. The others either showed less complementation (N28-mNG-L29), or slower growth rate (P12-mNG-A13, N28-mNG-L29, Q113-mNG-L114), or polar cell localization beside the midcell localization (Q113-mNG-L114, Q124-mNG-M125, Q151-mNG-T152, Q182-mNG-T183, Q212-mNG-T213, FtsN-mNG), or the tag is too close to the Essential domain of FtsN (K69-mNG-V70). For different imaging purposes, different tags or different fusions were used as indicated below.

In the 3D live-PALM imaging assay (Fig. 1b), a mEos3.2-FtsN fusion was used since a truly monomeric photoactivatable fluorescent protein mEos3.2[54] was needed for the single-molecule localization microscopy. Here mEos3.2 was fused to the N-terminus of FtsN.

In the FRAP assay (Fig. 1e), a GFP-FtsN fusion was used since GFP was easily photobleached, which contributes a very low background signal after the photobleaching. Here GFP was fused to the N-terminus of FtsN.

In the TIRF and TIRF-SIM assays (Fig. 1f), a mNG-FtsN fusion was used since mNG is much brighter and more photo-stable than GFP. Here mNG was fused to the N-terminus of FtsN.

In the SMT assay (Figs. 2–4), a $FtsN^{E60-E61}$-Halo (termed FtsN-$Halo^{SW}$) sandwich fusion was used since E61 was included in all the FtsN mutants ($FtsN^{Cyto-TM}$ is $FtsN^{1–73}$, $FtsN^{\Delta Cyto-TM}$ is $FtsN^{61–319}$, $FtsN^E$ is $FtsN^{61–105}$) used in the SMT imaging. Here Halo tag was either inserted between E60 and E61, or fused to E61 on FtsN mutants. There are two advantages to use the sandwich fusion rather than the N-terminal fusion in the SMT assays: (1) The Halo tag is in the same position on FtsN among WT FtsN and all FtsN mutants, which could eliminate the potential influence caused by different positioning of the tag; (2) E60-E61 is far away from key amino acids in FtsN (e.g., D5, W83, Y85, and Q251), which could lessen potential interferences.

The functionality of mEos3.2-FtsN, GFP-FtsN, mNG-FtsN, and FtsN-$Halo^{SW}$ fusions used in the imaging was tested by complementation, cell growth rate, cell length, and midcell localization (Supplementary Fig. 2 and Supplementary Fig. 3). Their stability and expression level were tested by Western blotting (Supplementary Fig. 4).

## 3D live cell SMLM imaging

The 3D live cell SMLM imaging was conducted on a home-built microscope as previously described[30]. Briefly, the green state of mEos3.2 before activation was excited at 488-nm (laser power 40 W cm$^{-2}$) to obtain integrated green fluorescence intensity of individual cells, which was used to quantify the percentage of FtsN-ring intensity in Fig. 1d. mEos3.2 was then photo-activated by using a 405-nm laser with intensity increased stepwise from 0 to 12 W cm$^{-2}$ to compensate for the gradually depleted pool of inactivated mEos3.2. The activated mEos3.2 was excited at 568-nm (laser power 1.6 kW cm$^{-2}$) continuously with a 10-ms exposure time.

3D imaging was achieved as previously described[30]. Briefly, a cylindrical lens (Thorlabs Inc) with 700-mm focal length was placed in the microscope emission pathway to introduce astigmatism to the single-molecule PSF[53]. TetraSpeck fluorescent microspheres with average diameter 0.1 μm (Invitrogen Molecular Probes) were used to calibrate the z-dependent changes to the shape of the astigmatic PSF. The xy positions were determined through the 2D Gaussian fitting of the PSF, while the z position was given by the calibration curve obtained by z-scanning of the fluorescent microspheres[53]. Because of the refractive index mismatch between the transmission path of the microspheres used for calibration (glass and oil) and that of the fluorescent proteins (aqueous cell environment, glass, and oil), the z values obtained from the calibration curve were rescaled by a factor of 0.75. The measurements of ring dimensions were achieved by custom MATLAB software described previously[27]. To quantitatively compare with previously reported dimensions obtained under different spatial resolutions, FtsN-ring dimensions here were deconvolved as described[27].

To quantitatively compare the distributions of clusters in FtsN- and FtsZ-rings, we used a previously established autocorrelation analysis[27,30]. In this analysis, all FtsN or FtsZ molecules in the ring are projected along the circumference of the ring. The spatial auto-correlation function (ACF) is calculated as the apparent probability distribution of linear distances between all molecule pairs along the circumference of the ring using the formula:

$$p(r_k) = \frac{\sum_{i=1}^{N-k} Z_i Z_{i+k}}{\sum_{i=1}^{N} Z_i^2} \quad k = 0, 1, \ldots, N-1$$

where $i$ is the index of the trajectory ($Z_1, Z_2, Z_3 \ldots Z_N$), $k$ is the distance lag between individual data points, and $N$ is the total number of data points in a trajectory.

The mean ACF curve of FtsN-rings had a significantly lower correlation value at short distances (Fig. 1c), suggesting that FtsN clusters are more homogenously distributed in FtsN-ring than those in the FtsZ-ring.

## Fluorescence recovery after photobleaching (FRAP)

FRAP experiment was performed on a home-built microscope as previously described[26,36]. Briefly, the excitation laser (488 nm) was split with the combination of a linear polarizer and a polarizing beam-splitting cube to generate a transmitted photobleaching beam and a reflected epifluorescence-illumination beam. The transmitted beam was focused to a diffraction-limited spot on one side of the FtsN-ring for photobleaching, while the reflected beam was used to image the cell before and after photobleaching. Images were acquired every 1 s for 150 s after photobleaching, with a 50-ms exposure time (Supplementary Movie 1). Custom MATLAB scripts as described previously[26] were used to analyze FRAP curves. The fluorescent intensity of the photobleaching area was normalized from 0 to 1, with the first acquisition right after photobleaching set as 0. The average intensity of the last 20 frames of the opposite side of the bleaching area in the same ring served as the maximum to normalize the intensity of the bleaching area (the maximum after normalization is 1 when the ring became homogenous after recovery). The global photobleaching was corrected by using the fluorescent intensity outside the septum. The FtsN FRAP curve presented in Fig. 1e was the average of two independent experiments (~20–40 cells in each experiment). The FtsZ FRAP curve is the data from a previous work[36]. The control data in Supplementary Fig. 7B was from the adjutant cells ($n = 6$) that were not photobleached (yellow arrowhead in Supplementary Fig. 7A set as an example). In the control, the fluorescent intensity of the first acquisition was very close to the rest since there was no photobleaching. The FRAP curve was close to 0 after subtracting the first acquisition.

The diffusion coefficient (D) of a typical inner membrane protein in prokaryotes is from 0.0075 to 0.22 $\mu m^2 s^{-1}$, depending on the protein size, protein surface charge, and number of transmembrane helices, etc.[82]. More specifically, in our recent work, we showed that the diffusion coefficients of three divisome proteins FtsI, PBP2b, and FtsW outside the septum in wildtype *E. coli*, *B. subtilis*, and *S. pneumoniae* were 0.041, 0.038, and 0.028 $\mu m^2 s^{-1}$, respectively[39]. The average unwrapped two-dimensional projected area of the septum in *E. coli* cells during division is ~0.2 $\mu m^2$ (600 nm in diameter and 100 nm in width on average). Half of the septal FtsN-ring was bleached in the FRAP experiment, producing a ~0.1 $\mu m^2$ bleaching area (A). Thus, the time that a random inner membrane divisome protein diffuses in and out of the bleaching area is ~2.5 s (took 0.04 $\mu m^2 s^{-1}$ as the D and calculated by A/D). The half times of the fast recovery phase of FtsN observed in this work (2.9 ± 0.8 s) and by Söderström et al. (1.87 ± 0.66 s)[41] are both very close to this time, indicating that the fast recovery phase was most likely contributed by the random diffusion of FtsN molecules in and out of the bleaching area at the septum.

## TIRF and TIRF-SIM imaging and data analysis

TIRF imaging was performed on a home-built microscope as previously described[36]. Briefly, the objective-based TIRF illumination was achieved by shifting the expanded 488-nm laser beam (Coherent Sapphire 488) off the optical axis center. The TIRF imaging angle was measured with a 20-mm right-angle prism (refractive index = 1.518, Thorlabs PS908) and fixed at ~70°. FtsN cluster dynamics were monitored by exciting the mNG-FtsN fusion strain at 488 nm (laser power 0.5 W cm$^{-2}$). The exposure time was set at 1 s. 200 frames were acquired continuously without any interval dark time.

TIRF-SIM imaging was performed on a General Electric (GE) Deltavision OMX-SR super-resolution microscope with a 60 × 1.49

UPlanApo oil objective and three high-speed high-sensitivity PCO sCMOS cameras to achieve higher temporal and spatial resolutions. The TIRF imaging angle was tuned from three directions by using the TetraSpeck fluorescent microspheres sample. The incident excitation power at 488-nm was adjusted to 6% transmittance (6.0% T) to minimize photobleaching. Time-lapse TIRF-SIM imaging was implemented with a 50-ms exposure time. 40 frames were acquired with 1 s interval dark time.

Cluster dynamics analysis was performed by using the ImageJ kymograph plugin (http://www.embl.de/eamnet/html/body_kymograph.html, J. Rietdorf and A. Seitz, EMBL, Heidelberg) as previously described[36]. Briefly, the fluorescence images of individual cells from TIRF or TIRF-SIM experiments were cropped, corrected for photobleaching, interpolated to 20 nm pixel$^{-1}$ via the bicubic method in ImageJ, and moving-averaged over a 4-frame window of time. The fluorescence intensity of an FtsN cluster along the direction of its movement in each frame was determined from the intensity along a line with a width of 11 interpolated pixels (~200 nm) manually drawn across the full length of the path of the FtsN cluster. This line was then used to plot the kymograph in Fig. 1f and Supplementary Fig. 8. The processively moving speeds of the cluster were calculated by manually measuring the slopes of the center line of the fluorescence zigzags in the kymograph. Kymographs without obvious fluorescence zigzags were not analyzed. The speed distribution of FtsN clusters (Fig. 1g) is the combination from TIRF and TIRF-SIM data.

The mean directional speed measured from the kymographs was at 8.8 ± 0.3 nm s$^{-1}$. The TIRF illumination region is approximal 500 nm in width according to a previous calculation[36]. Thus, the time that an FtsN cluster moves across the illumination region is ~57 s, which is essentially the same as the recovery half time of the slow phase (54 ± 10 s) we observed in the FRAP experiment, indicating that the directionally moving FtsN clusters are likely the ones contributing to the slow recovery rate of FRAP.

## Cell-labeling with Janelia Fluor 646 (JF646) dye

Cells from 1 ml of culture at an OD$_{600}$ ~ 0.35 were harvested by centrifugation and the cell pellet was resuspended with 1 ml M9$^-$ minimal medium or EZRDM including 1 nM JF646 (for SMT imaging) or 1 μM JF646 (for ensemble imaging). The culture was mixed well with a pipette and put on a nutator at RT for 30 min. After labeling, cells were washed three times with M9$^-$ minimal medium or EZRDM and concentrated to 50 μl.

## SMT imaging and data analysis

Cells were grown and labeled as described above except for certain conditions listed below. When cells were grown in EZRDM, the saturated culture was diluted 1:100 to fresh EZRDM medium with IPTG (and 0.2% L-arabinose for UppS induction) and allowed to grow at RT for 3 h to reach the log phase. For the fixed-cell control, log-phase cells were first labeled and then fixed as described previously[26]. Cells were then loaded onto a 50 μl, 3% agarose gel pad (containing the same growth medium without antibiotic) laid in an observation chamber (FCS2, Bioptechs). The chamber was locked on the microscope stage (ASI, Eugene, OR) to minimize mechanical drifts.

For drug-treated conditions, 0.5 μl of appropriate drug solution was added to the 50 μl concentrated labeled cells. The final concentrations used were: aztreonam 1 μg ml$^{-1}$, fosfomycin 200 μg ml$^{-1}$, and MTSES 100 μM. 0.5 μl of appropriate drug solution was also added on top of the gel pad right before applying cells. The moment cells were applied was counted as time zero. With MTSES treatment, the chamber with cells was kept on the microscope stage for 60 min before the images were acquired. With aztreonam treatment, the chamber with cells was kept on the microscope stage for 30 min before the images were acquired. With fosfomycin treatment, the chamber with cells was kept on the microscope stage for 30 min

before the images were acquired. All images were collected within 3 h of drug treatment.

SMT imaging was performed on an Olympus IX71 inverted microscope equipped with a 100×, 1.49 NA oil-immersion objective and Andor iXon 897 Ultra EM-CCD camera in epifluorescence-illumination mode using Metamorph 7.8.13.0 software. The focal plane was placed at ~250 nm from the bottom of the cell to image the molecules moving on the bottom half of the cylindrical portion of the cell body. Single molecules were tracked with 100 ms exposure time using a 647-nm laser with intensity 30 W cm$^{-2}$. The long exposure time helped to filter out molecules randomly diffusing along the cylindrical part of the cell body. 150 frames were acquired with 1 frame per second (1 fps). 3D imaging was achieved as described above.

The data processing was similar as previously described[39,40]. To specify, the $xy$ positions of single molecules were determined through the 2D Gaussian fitting of the PSF with ThunderSTORM[83], a plug-in for ImageJ[84], while the $z$ position was given by the calibration curve obtained by $z$-scanning of the fluorescent microspheres[53]. Because of the refractive index mismatch between the transmission path of the microspheres used for calibration (glass and oil) and that of the fluorescent proteins (aqueous cell environment, glass, and oil), the $z$ values obtained from the calibration curve were rescaled by a factor of 0.75. A bandpass filter (60–300 nm) for both sigma1 and sigma2 was applied to remove the single pixel noise and out-of-focus molecules. All analysis thereafter used custom scripts in MATLAB R2020a. The localizations were linked to trajectories using a home-built MATLAB script[85] that adopted the nearest neighbor algorithm from ref. [86]. The distance threshold was set to 300 nm per frame, which approximates to a max diffusion coefficient of ~0.05 μm$^2$ s$^{-1}$, or a max speed of 300 nm s$^{-1}$. To link molecules which may have blinked across frames or left the focal plane, a time threshold of 8 frames was chosen according to the off-time distribution. Only trajectories near the midpoint of the cell's long axis or near visible constriction sites where cell division takes place were used in the analysis to ensure the molecules are cell division and sPG related.

Due to the rod-shape cell envelope, the real displacement of the tracked molecules around the circumference is underestimated. The trajectories were unwrapped to one dimension using a home-built MATLAB script. We noticed that the velocity estimated from MSD curve fitting is not accurate when the dwell time of the trajectory is short (<20 frames) or when there is more than one moving state in a single trajectory. Unwrapped trajectories were then segmented manually to determine whether a single molecule in a segment is stationary or moving processively as previously described[40]. Briefly, a segment was first fitted with a line. $R$, which is the ratio of the displacement and the standard deviation of fitting residuals, and $P$, which is the probability of processive movement, were defined and used as the criteria for the classification of segments. Through manual inspection, we determined to classify segments as processive based on a threshold of $R \leq 0.4$ and $P \geq 0.5$, while all others were classified as stationary. Since the confidence of classification is correlated with the segmentation length, we only consider segments longer than 5 frames to minimize classification error.

The cumulative probability density (CDF) of directional moving FtsN speeds was calculated for each condition and fit to either a single or double log-normal populations:

$$CDF = P_1 \frac{\left(1 + \text{erf}\left[\frac{lnv - u_1}{\sqrt{2\sigma_1}}\right]\right)}{2} + (1 - P_1) \frac{\left(1 + \text{erf}\left[\frac{lnv - u_2}{\sqrt{2\sigma_2}}\right]\right)}{2}$$

where $v$ is the moving speed for FtsN or its mutants and $P_1$ is the percentage of the first population. For fitting with a single population, $P_1 = 1$. The values $u$ and $\sigma$ are the natural logarithmic mean and standard deviation. The average speed of each population is

calculated as $\exp(u + \frac{\sigma^2}{2})$. To estimate the error of the speed and percentage (Supplementary Tables S5), the CDF curves were boot-strapped 200 times and fit with the corresponding equation (single- or double-population).

To fit the stationary population, histograms of the velocities were generated from corresponding stationary segments in the respective FtsN condition data; the bins used in these histograms were the same for the final respective plots as in Figs. 1–4. Peak values and bin centers were used to fit a single exponential decay function:

$$f(x) = A * \exp(-\lambda x)$$

where $A$ is the amplitude of the fitted curve and $\lambda = \frac{1}{u}$. $u$ is the mean "velocity" of stationary segments.

### Reporting summary

Further information on research design is available in the Nature Research Reporting Summary linked to this article.

### Data availability

The authors declare that all data supporting the findings of this study are available within the paper and its supplementary information files. Source data are provided with this paper.

### Code availability

Code for analyzing single-molecule tracking data is available on the Xiao Lab GitHub repository[85].

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

## Acknowledgements

The authors thank all members of the Xiao and Weiss laboratories for helpful discussions and feedback on the manuscript, members of the Weiss lab for help with strain construction, the Microscopy Facility of Johns Hopkins School of Medicine for assistance with the TIRF-SIM imaging, and Dr. L. Lavis for sharing JF549 and JF646. Work in the Xiao lab was supported by NIH R01GM086447 and R35GM136436 (to J.X.), GM125656 (subcontract to J.X.), a Hamilton Smith Innovative Research Award (to J.X.), and in part by NIH GM007445 (to J.W.M.). Work in the Weiss lab was supported by NIH R01GM125656 (to D.S.W.) and in part by T32AI007511 to G.M.K.

## Author contributions

Z.L., D.S.W., and J.X. conceived the study. A.Y., G.M.K., D.S.W., and Z.L. constructed the strains and performed genetic and phenotypic experiments. Z.L. performed all the imaging experiments and analyzed the data. X.Y. and J.W.M. wrote the custom MATLAB script for analyzing single-molecule tracking data. Z.L. analyzed the single-molecule tracking data with help from X.Y., J.W.M. and R.M. Z.L., D.S.W.. and J.X. wrote the original draft. All authors reviewed and edited the manuscript. D.S.W. and J.X. supervised the study. Funding was acquired by D.S.W. and J.X.

## Competing interests
The authors declare no competing interests.
