## [Peer Review File · Nature Communications]

Reviewer Comments, first round

Reviewer #1 (Remarks to the Author):

The manuscript by Lyu and colleagues describes the dynamics of FtsN at the septum, showing that it moves processively at a speed similar to that of FtsWI molecules. This movement is not dependent on FtsZ treadmilling, but is driven exclusively by peptidoglycan synthesis. The role of FtsN and the mechanisms by which it activates FtsWI has been a matter of study for years, and this manuscript contributes to its clarification.

This is a clear, well-written manuscript, describing carefully executed experiments in a topic relevant to the field.

Major comments

General comments: Different FtsN fusions to fluorescent proteins or Halo Tag were used for this work. The rationale for the use of different fusions is clearly explained only in the supplementary material. It is therefore difficult for the reader, while reading the main text, to know exactly which fusion was used for which experiment and why. Legends of every panel should mention which fusion was used for that specific experiment (for example, in Fig 1 panels D, F, G have no indication of the fusion used). Also, the first time a new fusion is used, a brief explanation should be given in the main text. It is also difficult to understand for each experiment if native FtsN is being expressed (or not) together with the fluorescent derivative of FtsN, either due to IPTG induction or leakiness of the promoter. Again, growth conditions are given in the methods, but it would make the reader's life easier if a brief comment on the presence of native FtsN was given in the main text or figure legends.

Finally, the number of datapoints (n) is lacking for some experiments in the figure legends.

Line 25-26: It is stated in the abstract "Here we use single-molecule tracking to investigate how FtsN activates sPG synthesis in *E. coli*". This should be rephrased as the manuscript does not address directly the molecular mechanism of FtsWI activation by FtsN.

Line 114: Sup table 4 is not mentioned in the text, but it is a useful table that should be mentioned in this paragraph.

Lines 137-142: This data is in agreement with data from B. Söderström (ref 48) who clearly showed that FtsZ and FtsN rings do not overlap, with the FtsN ring being larger than the FtsZ ring during part of the cell cycle (Fig 4 of ref 48). The data in Fig 1D from this manuscript is less clear than that published data: Authors state that the FtsN ring disassembles at a ring diameter of ~ 300 nm. However, in Fig 1D, there is only a small drop in localization percentages in the plot for FtsN between the 4th and 5th datapoint ($\sim 3\%$?), which is described as disassembly of the ring, while for FtsZ this drop is $\sim 15\%$. In fact a similar drop of $\sim 3\%$ in FtsN localization percentages can also be observed between the 1st and 2nd datapoints, but it is not interpreted by the authors as disassembly. Also, please clarify the exact meaning of "midcell localization percentages" in line 727 (legend of Fig 1D). Is it the number of cells with FtsZ or FtsN localized at midcell, or the % of fluorescence corresponding to FtsZ or FtsN at midcell, versus the whole cell? If the latter, as I assume, please clarify.

Line 161-162: "They suggest that the spatiotemporal organization and dynamics of FtsN are most likely independent of FtsZ". Please rephrase as most likely authors do not mean FtsN organization is totally independent of FtsZ, as depletion of FtsZ would affect FtsN localization.

Line 195: At this point in the manuscript, it is not clear for the reader why authors change from an N-terminal fusion to the sandwich fusion. A brief explanation should be given in this paragraph.

Line 255: Is there any native FtsN expressed due to promoter leakiness in strains EC5263 and 5271 (Although pBAD is supposed to be very tight, was this checked by western?). Can authors state how much longer than WT are these cells?

Line 275 - If the SPOR domain is responsible for the static FtsN molecules, as it anchors FtsN to denuded glycans (lines 410-411 of discussion), how does the FtsN cyto-TM construct, which lacks the SPOR domain, have a large fraction of stationary molecules (Fig 2J)? This construct is expressed in the presence of WT FtsN (line 276). Can the FtsN cyto-TM interact with stationary WT FtsN molecules? But if that is the case, analysis of FtsN cyto-TM dynamics should be interpreted taking that in consideration.

Lines 384-393: A new, fast moving population emerges for FtsNE, that closely resembles the fast moving FtsW on the Z track. In order to make this conclusion, authors should evaluate the speed of this fast moving population of FtsNE in strains expressing FtsZ GTPase mutants with altered treadmilling speed, similarly to what was done for FtsN Cyto-TM in Fig 2J. Also, similarly to the comment above, why does the FtsNE construct, which lacks the SPOR domain, have a large fraction of stationary molecules?

Figure 1

Panel B – If FtsN localizes mainly at midcell, why is there so much signal in the cell periphery, even in cells that have a septum?

Panel E – data is the same as Fig S6B, which includes also the control. If the control is included in figure 1E, then Fig S6B can be removed. Also, please explain why the normalized intensity of the control is 0 and not 1.

Figure 2 – The x axis scale in panels E and J is different. This may mislead the reader when comparing the histograms for FtsN WT and FtsN cyto-TM.

Figure 2, panel H. This panel contains two sets of data which make it unnecessarily confusing. Please separate.

Figure 3, panel A- A MTSES control (added to BW25113 WT) should be included.

Figure 3, panel B – Y axis is not the same in three graphs, which is misleading for a reader who does not notice that.

Figure 3, panel D and sup Table 10 – BW25113 WT should be tested in EZRDM for comparison with M9-glucose.

Line 334 and Sup Table 10 – The increase of FtsN speed upon overexpression of UppS is minor (from 12.3 to 13.7 nm/s). This should be stated in the text.

Sup Fig 2 B – Was IPTG added and if so at what concentration?

Sup Fig 7 – Do not place scale bar along time. Also, visually the kymograph in the left, in Fig A iii which corresponds to a moving molecule is not very different from the kymograph in panel B which corresponds to a molecule that is not moving as it is in a fixed cell.

Methods - Description of plasmid pJL136 is missing

Minor comments

General - mNG is a more common abbreviation for mNeonGreen than mNeG

Line 103-104: delete "activity" in "activating sPG synthesis activity"

Line 154: replace "which is" at the end of the sentence by "and is"

Line 290 – typo in "stationary"

Page 9, SI, below equations – "its mutants", not "it's mutants"

Sup Fig 8 – state meaning of CDF in legend

Reviewer #2 (Remarks to the Author):

The manuscript by Lyu et al. reports on single-molecule studies of FtsN molecules in live *E. coli* cells. It has been proposed in the past that FtsN is the trigger for the onset of constriction in *E. coli*. The authors find that midcell FtsN proteins can be divided between two groups based on their speeds. One group consists of stationary molecules while the other of slow-moving molecules with speeds of about 10 nm/s. The speed of slow-moving FtsN is dependent on the septal peptidoglycan synthesis but not on FtsZ treadmilling. The earlier studies by the same group have shown that FtsW molecules (septal peptidoglycan transglycosylase) also show a population that moves at speeds about 10 nm/s, which are involved in septal peptidoglycan synthesis (septal peptidoglycan track). Furthermore, the same studies also showed a different group of FtsW moving at speeds 20-30 nm/s, which corresponds to the speed of FtsZ treadmilling. Based on these data, the authors propose that FtsN activates septal cell wall synthesis by capturing or retaining FtsWI on the septal peptidoglycan track.

The reported experiments are carefully done and reported findings appear overall solid with few exceptions. However, the claim that FtsN activates/triggers septal peptidoglycan synthesis is vague and not backed up by data. Although the work lacks new evidence that FtsN activates the septal cell wall synthesis, the presented data provide valuable insights on how FtsN is involved in cell division. These findings are of broad interest to people who study bacterial cell division and cell cycle.

Main points of criticism:

1) The main claim that FtsN activates/triggers septal peptidoglycan synthesis is vague and not backed up by data. The authors just show that FtsN likely moves together with the FtsI-FtsW. More careful wording of the title and conclusion in the abstract (lines 31-34) and elsewhere is warranted.

2) There is inconsistency in the claims on how FtsN is recruited to the septum. On one hand, the authors claim that FtsN activates sPG synthesis by capturing or retaining FtsWI on the sPG-track (abstract). However, in the main summarizing Figure (Fig. 4C) they show that FtsA recruits FtsN although the authors do not see this group of FtsN in their experiments, presumably because the period to observe this group is too transitory. But if FtsN indeed were the trigger, then this triggering would occur during this transitory period. The latter makes the point that the authors cannot claim that FtsN triggers the septal peptidoglycan synthesis as they are not observing it.

3) Why is there a population of stationary FtsN Cyto-TM molecules in Fig. 2J? How is Fig. 2I consistent with the explanation that stationary population corresponds to FtsN that is bound to sPG via its SPOR domain?
How is it ruled out the stationary population is not an analysis artifact? See also note to Lines 200-203 below.

4) The summary model in Fig. 4C is not consistent with the observations and previous reports:

1. There is no evidence of FtsN in the Z-track (rightmost part of the Figure). This should be observable in the experiments if, as the authors propose "FtsN is first recruited to the septum through the interaction between its cytoplasmic tail with FtsA, and is distributed around the septum by treadmilling FtsZ polymers." The distribution process should presumably take some time (about 1 minute to traverse half a perimeter).

2. It is not clear that the stationary population corresponds to FtsN that is bound to sPG via its SPOR domain because FtsN Cyto-TM shows a stationary population.

3. The current model supported by different groups suggests that one of the players in activating septal peptidoglycan synthesis is FtsQLB as has also been shown by authors in Figure 1A. However, the model in Figure 4C does not involve FtsQLB. None of the results in the manuscript show that FtsQLB complex is involved.

Minor points of criticism:

1) Lines 133-134: "However, autocorrelation analysis showed that the FtsN molecules in the FtsN-ring are more homogeneously distributed than those in the FtsZ-ring (Fig. 1C)". Fig.1C shows not an autocorrelation function but a pair-correlation function. The authors should specify if the distance r is a 3D distance or the distance on the cylindrical surface.

2) Fig 1B: The radial thickness of the FtsN-ring is listed 51 ± 4 nm. It is not feasible that the position of mEos3.2 can vary more than a few nanometers from the inner membrane because FtsN is an integral membrane protein. The listed number is rather experimental uncertainty. The same seems to apply also to the thickness of the FtsZ-ring because the numbers are comparable to that of the FtsN-ring.

3) Lines 138-141: "We found that FtsN-rings assemble at a ring diameter of ~ 600 nm and disassemble at ~ 300 nm (Fig. 1D). In contrast, under the same experimental condition FtsZ-rings assemble at ~ 950 nm and start to disassemble at ~ 600 nm (Fig. 1D)." These numbers are not consistent. FtsZ- and FtsN rings should assemble at the same diameter if FtsN "activates" septal cell wall synthesis. Also, 950 nm for *E. coli* diameter is too large.

4) Lines 197-198: "The Halo tag is inserted after amino acid E60, between the TM and E domains in the periplasm". Please explain why the Halo tag was inserted into this region instead of the cytoplasmic domain as in the measurements described earlier.

5) Lines 198-199: "We tracked septum-localized single FtsN-HaloSW molecules using a frame rate of 1 Hz to effectively filter out fast, randomly diffusing molecules along the cylindrical part of the cell body." I do not think the frame rate here is a relevant number to quote. One should instead mention the exposure time.

6) Lines 200-203: "Using a custom-developed unwrapping algorithm, 47, we decomposed 3D trajectories of individual FtsN molecules obtained from the curved cell surfaces at midcell to one-dimensional (1D) trajectories along the circumference and long axis of the cell respectively as previously described 47" The unwrapping algorithm is very sensitive to accurate determination of cell contours. The authors should explain how cell contours were determined and how the shift between brightfield and fluorescent images was handled. This information is missing from Ref 47.

As control of their method, the authors should plot the speed of FtsN-Halosw versus the radial distance of the molecule from the cell center.

7) Fig. 2H: the legend overshadows the data.

8) Lines 254-256: "Both mutant fusions were able to support cell division as the sole cellular FtsN and showed prominent midcell localization, but cells were both longer than WT ones (Fig. 2F and Supplementary Fig. 10)". Please mention in the text how much longer.

9) Figure 4A: Are FtsN WT cells shown here already with superfission variant ftsBE56A Δ ftsN background? If yes, then add this information to the Figure legend.

10) Lines 391-393: "In other words, the SPOR domain may be the major determinant to prevent the release of the sPG synthesis complex from the sPG-track to the Z-track." Is there any model that proposes that sPG synthesis complex is released from the sPG-track to the Z-track? What would be the function of such a release?

11) Lines 401-402: "In principle FtsN might localize to the Z-track and prevent or even disrupt binding of FtsWI to the Z-track." How is this statement consistent with the conclusion from the authors' previous paper that "FtsN promotes FtsW release from the Z-track to become active in sPG synthesis on the slow 'sPG-track'"?

12) Line 447: "we found that abrogating the FtsN-FtsA interaction, either by deleting the cytoplasmic domain or introducing a D5N substitution, resulted in mild cell elongation". This is misleading according to data shown on SI Fig.10, which shows that cell length increase 2x and 4x.

13) Lines 457-460: "In this model, FtsN is first recruited to the septum through the interaction between its cytoplasmic tail with FtsA, and is distributed around the septum by treadmilling FtsZ polymers. This period may be too transitory for us to observe a significant population of fast-moving, full length FtsN molecules in our experiments." If FtsN is appreciably spread over the perimeter of the cell by treadmilling then at speed 25 nm/s and should be observable. Some further explanation is warranted.

Reviewer #3 (Remarks to the Author):

The manuscript by Lyu et al. studies the dynamic behavior of the essential cell division protein FtsN in *E. coli*. FtsN has at three distinct domains: an N-terminal cytoplasmic peptide that interacts with FtsA, the essential E-domain in the periplasm that is known to interact with and activate FtsIW and the SPOR-domain that binds to denuded peptidoglycan strands. While it seems clear that all these interactions are involved in the recruitment of FtsN towards the division septum, it's not known how they individually contribute to control the timing of recruitment starting from the initial assembly of the Z-ring to the constriction of the cell septum.

From previous work we already know the interaction with the PG synthesis machinery is dominating during constriction, as there is a clear spatial separation between the FtsN and FtsZ. In contrast, FtsN remained colocalized with FtsI (Söderström et al *Molecular Microbiology* 2018). This study also found that the movements of FtsZ and FtsN are mechanistically different as demonstrated by their spatial separation and different fluorescence recoveries in FRAP experiments. Together, this study concluded that the FtsZ and FtsN are part of two distinct protein complexes that are becoming spatially separated during constriction.

The idea of separated complexes was picked up by another recent study from the group of Jie Xiao (Yang et al. *Nature Microbiology* 2021). Here, single molecule imaging helped to identify two tracks that gave rise to different dynamics of the septal cell wall synthesis complex FtsWI: first, a fast Z-track of inactive enzymes whose movement is driven by FtsZ treadmilling dynamics and second, a slow PG-track, where proteins move slowly powered by PG synthesis. This study found that FtsN activates FtsW and promotes its switch from fast to slow motion.

From these two studies, it is already clear that FtsN is first recruited to the septum via an interaction with FtsA but then forms a complex with FtsWI and PG. In contrast, there is no existing evidence for an alternative model, where FtsN would remain part of the Z-track during the entire process of cell division.

In the current paper, the authors analyze the single molecule behavior of FtsN. Consistent with the study by Söderström et al and Yang et al, they find that FtsN shows two different populations, an immobile and a moving one. In the later population, FtsN moves at a similar velocity as FtsW suggesting that its dynamics are not powered by FtsZ treadmilling, but by PG synthesis. This again confirms a model where FtsN predominantly localizes to the PG synthesis machinery.

Next, the authors construct a couple of different FtsN truncations and find that the dynamics of the N-terminal peptide are defined by its interaction with the Z-ring and that the interaction with FtsWI can drive the dynamics of the E-domain. These are interesting observations but again confirm already existing studies (Busiek et al 2012, 2014, Yahashiri et al 2017).

Overall, I think that studying the intracellular dynamics of cell division proteins is a powerful approach that offers new mechanistic insight, however, this paper does not provide any new information regarding the function and properties of FtsN. My biggest disappointment is that single molecule imaging of FtsN should in principle provide a wealth of data about its behavior during cell division but the authors do not take advantage of it. For example, from the velocity histograms of single molecule trajectories it is obvious that FtsNs exist in different complexes. From the single molecule trajectories, it should be possible to obtain information about the lifetime of these complexes and the respective switching rates between them. These numbers should change during cell division, i.e. be significantly different from the early stages of Z-ring assembly towards cell separation and should be very different for the different truncations. This kind of information is already would be incredibly useful to reveal a much more detailed picture at how FtsN is recruited to midcell and when and how it then moves on to activate FtsWI.

In addition, it is disappointing that the manuscript does not include any single molecule data (time lapse micrographs) that could support their findings and also include previously published data without properly disclosing it.

In summary, due to the lack of new observations, missing analysis and data, I cannot recommend publication of the manuscript.

Please find my more specific comments below:

Fig. 1:

This figure does not show new information, instead it is a confirmation of findings by Söderström et al. 2018. This should be properly discussed.

- Panel B contains data from a previously published manuscript, this should be clearly disclosed in the main text and figure caption.

- N-terminal mEos3.2-FtsN is missing in Supplementary Table 4. FtsN localization is not confined to midcell (compare with immunostainings in supplement). Immunostaining for mEos3.2 FtsN to confirm correct localization

- Can the authors do a significance test to demonstrate the difference between the autocorrelation profile? To me they do look similar. If the authors believe this analysis is useful, it should be done at different time points during division and for different versions of FtsN, i.e. is it still different for FtsN Cyto TM? What about FtsN delta Cyto TM?

- Panel C and D is a less detailed characterization of the spatial separation of FtsZ and FtsN observed before, it does not include new information.

- Incomplete FRAP recovery also due to limited amount of FtsN molecules in the living it is no indication of a distinct FtsN species. The supporting movie is not convincing.

Fig. 2:

- What is the lifetime of stationary complexes? Denuded Glycans are only transiently present in SPG, this could be measured by getting a histogram of stationary lifetimes.
- Line 199: 1Hz framerate is too slow to visualize fast diffusive behavior and short-lived states, which would be interesting to study to understand the behavior of FtsN. It is also incorrect that a long frame rate filters out random diffusion, long exposure time does this.
- Where are there time lapse movies of single molecule experiments?
- Please use better color coding, some plots are difficult to understand (Fig. 2H and 2K for example)
- What can explain the immobile fraction of the different FtsN truncations? While it is straightforward to explain this for the full-length protein, it's not clear how for FtsN Cyto TM.
- Line 290: "The rest FtsN segments..." something is missing in this sentence
- Instead of using FtsN D5N, could the authors use an N-terminal truncation?

Fig. 3:

- What explains the stationary protein?
- Or is it possible to de- or increase the presence of denuded peptidoglycan to see if this stationary fraction is depending on the denuded PG?

We thank the reviewers for their constructive comments. Below we provide point-to-point responses (blue) to reviewers' comments (black). To facilitate the reading of the revised manuscript, we highlighted major changes in blue in both the main text and supplemental information. Small typos and grammar corrections are not highlighted.

Reviewer #1 (Remarks to the Author):

The manuscript by Lyu and colleagues describes the dynamics of FtsN at the septum, showing that it moves processively at a speed similar to that of FtsWI molecules. This movement is not dependent on FtsZ treadmilling, but is driven exclusively by peptidoglycan synthesis. The role of FtsN and the mechanisms by which it activates FtsWI has been a matter of study for year, and this manuscript contributes to its clarification. This is a clear, well written manuscript, describing carefully executed experiments in a topic relevant to the field.

Thank you!

Major comments

General comments: Different FtsN fusions to fluorescent proteins or Halo Tag were used for this work. The rationale for the use of different fusions is clearly explained only in the supplementary material. It is therefore difficult for the reader, while reading the main text, to know exactly which fusion was used for which experiment and why. Legends of every panel should mention which fusion was used for that specific experiment (for example, in Fig 1 panels D, F, G have no indication of the fusion used). Also, the first time a new fusion is used, a brief explanation should be given in the main text. It is also difficult to understand for each experiment if native FtsN is being expressed (or not) together with the fluorescent derivative of FtsN, either due to IPTG induction or leakiness of the promoter. Again, growth conditions are given in the methods, but it would make the reader's life easier if a brief comment on the presence of native FtsN was given in the main text or figure legends. Finally, the number of datapoints (n=) is lacking for some experiments in the figure legends.

Thank you for these suggestions. We have revised the manuscript as follows.

1. Strain numbers are now mentioned in the text when the strains are used.
2. Growth conditions are the same in almost all experiments, so they are described early in the Results section (lines 113-118) as follows: "*Except where stated otherwise, all experiments described below used cells grown in M9-glucose minimal media supplemented with IPTG in the absence of arabinose. Under these conditions, the fluorescent FtsN fusion protein is the only FtsN in the cells (Supplementary Fig. 4). Expression, stability and functionality of the fusions were validated by Western blotting and cell growth measurements (Supplementary Fig. 4 and Supplementary Table 4).*"
3. A description of each fusion and why it was chosen has been integrated into the text when that fusion is first used.
4. The number of data points was added in the figure legends.

Line 25-26: It is stated in the abstract "Here we use single-molecule tracking to investigate how FtsN activates sPG synthesis in *E. coli*". This should be rephrased as the manuscript does not address directly the molecular mechanism of FtsWI activation by FtsN.

We changed the sentence to "*Here we use single-molecule tracking to investigate FtsN's dynamics during sPG synthesis in E. coli.*"

Line 114: Sup table 4 is not mentioned in the text, but it is a useful table that should be mentioned in this paragraph.

Now mentioned in lines 101 and 118.

Lines 137-142: This data is in agreement with data from B. Söderström (ref 48) who clearly showed that FtsZ and FtsN rings do not overlap, with the FtsN ring being larger than the FtsZ ring during part of the cell cycle (Fig 4 of ref 48). The data in Fig 1D from this manuscript is less clear than that published data: Authors state that the FtsN ring disassembles at a ring diameter of ~300nm. However, in Fig 1D, there only a small drop in localization percentages in the plot for FtsN between the 4th and 5th datapoint (~3%?), which is described as disassembly of the ring, while for FtsZ this drop is ~15%. In fact a similar drop of ~3% in FtsN localization percentages can also be observed between the 1st and 2nd datapoints, but it is not interpreted by the authors as disassembly.

Also, please clarify the exact meaning of “midcell localization percentages” in line 727 (legend of Fig 1D). Is it the number of cells with FtsZ or FtsN localized at midcell, or the % of fluorescence corresponding to FtsZ or FtsN at midcell, versus the whole cell? If the later, as I assume, please clarify.

The midcell localization percentage is the “% of fluorescence corresponding to FtsZ or FtsN at midcell, versus the whole cell”, as the reviewer correctly assumed. We added a description in the main text to clarify the definition (lines 137-138). We also revised the text to reflect the possibility that at ~ 300 nm the ~ 3% drop in FtsN’s midcell localization likely indicates that from 600 to 300 nm, the FtsN ring does not disassemble significantly, in contrast to the FtsZ ring. See lines 130-147 and also copied below:

“We observed that FtsN rings are patchy (Fig. 1B) as previously reported, and that FtsN exhibits significant membrane localization along the perimeter of the cell. The high spatial resolutions (~ 50 nm in xy and ~ 80 nm in z, Supplementary Fig. 6) revealed that FtsN-rings have a comparable width and thickness to FtsZ-rings^{27, 29, 30, 32, 34} (Fig. 1B, Supplementary Fig. 6C, and Supplementary Table 5) and that FtsN molecules in the FtsN-ring are more homogeneously distributed than FtsZ molecules in the FtsZ-ring, as indicated by the autocorrelation analysis (Fig. 1C). To explore how FtsN-rings assemble and disassemble during cell division, we calculated the midcell localization percentages of FtsN by dividing midcell ring fluorescence by the whole cell fluorescence. We observed that maximally ~20% of cellular FtsN molecules accumulated in the FtsN ring (Fig. 1D). This value is in agreement with a recent study using a fluorescent ftsN fusion expressed from ftsN’s native chromosomal locus⁵⁵. In contrast, the FtsZ ring contained up to ~45% of the cellular pool of FtsZ (Fig. 1D). When cells of different ring diameters were arranged to generate a pseudo time lapse representing the cell wall constriction process, we observed that FtsN-rings assembled at a ring diameter of ~ 600 nm and only disassembled modestly through ~ 300 nm (Fig. 1D). In contrast, under the same experimental condition FtsZ-rings assembled at ~ 950 nm and started to disassemble drastically at ~ 600 nm (Fig. 1D).”

Line 161-162: “They suggest that the spatiotemporal organization and dynamics of FtsN are most likely independent of FtsZ”. Please rephrase as most likely authors do not mean FtsN organization is totally independent of FtsZ, as depletion of FtsZ would affect FtsN localization.

We revised to text to be “different” instead of “independent” of FtsZ. See lines 147-148 and copied here:

“These observations are consistent with previous evidence that FtsN and FtsZ rings have different spatiotemporal organizations⁴¹.”

Line 195: At this point in the manuscript, it is not clear for the reader why authors change from an N-terminal fusion to the sandwich fusion. A brief explanation should be given in this paragraph.

We added lines 205-208: *“Here we switched from N-terminal fusions to a sandwich fusion because we could use the same Halo insertion site when comparing the dynamics of full-length FtsN to those of FtsN deletion derivatives that lack the cytoplasmic or periplasmic domain.”*

Line 255: Is there any native FtsN expressed due to promoter leakiness in strains EC5263 and 5271 (Although pBAD is supposed to be very tight, was this checked by western?). Can author state how much longer than WT are these cells?

We checked the expression by Western and found that promoter leakiness is not an issue. We also measured the cell length. We expanded the description of these two mutants in lines 265-271: *“Both mutant fusions were produced in a P_{BAD}::ftsN depletion strain grown in M9-glucose plus IPTG. Western blotting verified that native FtsN was effectively depleted and the fusions were produced at physiologically appropriate levels (Supplementary Fig. 11). Both mutant fusion proteins showed prominent midcell localization and supported cell division, but cells were about twice as long as WT (Fig. 2F and Supplementary Fig. 11), likely due to delayed septum localization of FtsN and/or slowed rate of cell wall constriction.”*

Line 275 - If the SPOR domain is responsible for the static FtsN molecules, as it anchors FtsN to denuded glycans (lines 410-411 of discussion), how does the FtsN cyto-TM construct, which lacks the SPOR domain, have a large fraction of stationary molecules (Fig 2J)? This construct is expressed in the presence of WT FtsN (line 276). Can the FtsN cyto-TM interact with stationary WT FtsN molecules? But if that is the case, analysis of FtsN cyto-TM dynamics should be interpreted taking that in consideration.

We do not believe that FtsN’s Cyto-TM domain could interact with stationary WT FtsN molecules through the Cyto-TM domain, because the FtsN^{Cyto-TM-D5N}-Halo fusion (contains the D5N point mutation interrupting interactions with FtsA) completely abolished any residual midcell localization, indicating that here the midcell localization is mediated by FtsA.

Stationary FtsN^{Cyto-TM}-Halo molecules are most likely due to the interaction with FtsA, which can bind stationary FtsZ subunits in the middle of treadmilling FtsZ polymers. We have reported a similar stationary population of FtsW in our previous work. Consistent with this possibility, the average lifetime of these stationary FtsN^{Cyto-TM}-Halo^{SW} molecules is in the range of 12- 19 s (Supplementary Table 9), significantly shorter than that of the full length FtsN-Halo^{SW} fusion (~ 30 s), but similar to that of stationary FtsW molecules and FtsZ subunits under the same growth and imaging conditions. Please see lines 309-315 for the explanations.

Lines 384-393: A new, fast moving population emerges for FtsNE, that closely resembles the fast moving FtsW on the Z track. In order to make this conclusion, authors should evaluate the speed of this fast moving population of FtsNE in strains expressing FtsZ GTPase mutants with altered treadmilling speed, similarly to what was done for FtsN Cyto-TM in Fig 2J. Also, similarly to the

comment above, why does the FtsNE construct, which lacks the SPOR domain, have a large fraction of stationary molecules?

Following this suggestion, we have indeed tried this experiment. However, despite our best effort, Halo-FtsN^E fusion showed abnormal and static polar localization in FtsZ WT and mutant backgrounds, and hence we were unable to track their dynamics at the septum. It was unclear whether this polar localization was caused by the Halo fusion, or the poor localization of the FtsN^E fragment in the absence of the SPOR domain and the N-terminus. We observed clear midcell localization of FtsN^E only in the superfiSSION *ftsB*^{E56A} background, which could boost the interaction between FtsN^E fragment and FtsWI. Indeed, when we constructed two FtsZ GTPase mutants (E250A and G105S) in the *ftsB*^{E56A} background, we observed exactly the expected behavior. See lines 408 to 416:

*“Importantly, we further confirmed that the fast-moving population of Halo-FtsN^E is indeed on the Z-track by showing that its velocity was reduced in two FtsZ GTPase mutants with diminished treadmilling dynamics (*ftsZ*^{E250A}, *ftsZ*^{G105S}, Fig. 4C, Supplementary Fig. 14B, Supplementary Table 12). Both the percentage and mean dwell time of stationary FtsN^E molecules increased with reduced FtsZ GTPase activity (Supplementary Fig. 15, Supplementary Table 12), in line with the increased FtsZ filament length in FtsZ GTPase mutants, and similar to those of the slow-moving population of FtsW under identical conditions⁴⁰.”*

Similar to what we described above, because the FtsN^E fragment is in complex with FtsWI, it follows that FtsN^E has a stationary population as FtsWI—Supplementary Table 12 shows that the lifetime of the stationary population of FtsN^E (~ 14 s) is on par with that of FtsWI we previously measured.

Figure 1 Panel B – If FtsN localizes mainly at midcell, why is there so much signal in the cell periphery, even in cells that have a septum?

Ours (Fig. 1D) and other’s (Mannik *et al.*, bioRxiv, 2021) measurements both showed that at maximum only ~20% of FtsN molecules localize at midcell to form a ring. FtsN is a membrane protein, which diffuses at ~ 0.04 $\mu\text{m}^2 \text{s}^{-1}$ when it’s outside the midcell. Within the exposure time used in SMLM imaging (10 ms), single FtsN molecules diffuse within an area of ~ 0.4 $\times 10^{-3} \mu\text{m}^2$, which is much smaller than the xy- 50 nm detection resolution area (50 nm \times 50 nm = 2.5 $\times 10^{-3} \mu\text{m}^2$), and thus could be detected in the periphery membrane and localized as single molecules in SMLM imaging. In contrast, FtsZ is a cytoplasmic protein which diffuses at 0.75 $\mu\text{m}^2 \text{s}^{-1}$ when not forming a filament ([https://www.cell.com/biophysj/pdf/S0006-3495\(10\)04142-1.pdf](https://www.cell.com/biophysj/pdf/S0006-3495(10)04142-1.pdf)). Within the 10 ms exposure time, single FtsZ molecules diffuse within an area of 7.5 $\times 10^{-3} \mu\text{m}^2$, which would be blurred out in the imaging and cannot be localized as single molecules in the cytoplasm. All the divisome proteins (including FtsN and FtsZ) that localize to the midcell, however, have much smaller diffusion coefficients and thus could be detected and localized in SMLM imaging, showing prominent midcell intensity.

Panel E – data is the same as Fig S6B, which includes also the control. If the control is included in figure 1E, then Fig S6B can be removed. Also, please explain why the normalized intensity of the control is 0 and not 1.

We removed Fig. 1E data from Fig. S6B (now is Fig. S7B) and added lines 225-231 in the supplementary note: *“The fluorescent intensity of the photobleaching area was normalized from 0 to 1, with the first acquisition right after photobleaching set as 0. The average intensity of the last 20 frames of the opposite side of the bleaching area in the same ring served as the maximum*

to normalize the intensity of the bleaching area (the maximum after normalization is 1 when the ring became homogenous after recovery). The global photobleaching was corrected by using the fluorescent intensity outside the septum.” In the control, the fluorescent intensity of the first acquisition was very close to the rest since there was no photobleaching. The FRAP curve was close to 0 after subtracting the first acquisition.

Figure 2 – The x axis scale in panels E and J is different. This may mislead the reader when comparing the histograms for FtsN WT and FtsN cyto-TM.

We decided not to make the x axes of E and J the same because the two x axes are drastically different: E is 0-20 nm/s, and J is 0-60 nm/s. We would need three-fold larger space to accommodate the two panels if the x axes were the same.

Figure 2, panel H. This panel contains two sets of data which make it unnecessarily confusing. Please separate.

Panel H and K used two y-axes to accommodate the limited space to illustrate both the percentage of the moving population and the average moving speed. We used a smaller legend font size in the new Fig. 2H to make it look cleaner.

Figure 3, panel A- A MTSES control (added to BW25113 WT) should be included.

We have done this control and the data is now included in Supplementary Fig. 12. The BW25113 with MTSES control showed essentially the same moving dynamics of FtsN as that without MTSES.

Figure 3, panel B – Y axis is not the same in three graphs, which is misleading for a reader who does not notice that.

We decided not to make this change because the bin numbers are different for these histograms due to different sample sizes.

Figure 3, panel D and sup Table 10 – BW25113 WT should be tested in EZRDM for comparison with M9-glucose.

We have done this experiment but did not see a dramatic change compared to what we observed in the superfission mutant background. It is possible that the EZRDM and the M9-glucose medium provide similar levels of precursor so that there is no significant enhancement of the sPG activity of the WT FtsWI complex.

Line 334 and Sup Table 10 – The increase of FtsN speed upon overexpression of UppS is minor (from 12.3 to 13.7 nm/s). This should be stated in the text.

We believe that the relatively modest acceleration of FtsN speed from EZRDM to UPPS overexpression reflects the fact that we are approaching the maximal polymerization rate by FtsW under such conditions.

Sup Fig 2 B – Was IPTG added and if so at what concentration?

No IPTG was used. We added this information in the figure legend.

Sup Fig 7 – Do not place scale bar along time. Also, visually the kymograph in the left, in Fig A iii which corresponds to a moving molecule is not very different from the kymograph in panel B which corresponds to a molecule that is not moving as it is in a fixed cell.

We moved the yellow scale bars to the bottom of the kymographs. Another representative cell was chosen to replace the current one in Fig. 8Aiii.

Methods - Description of plasmid pJL136 is missing

We have added this description. See Supplementary Information lines 838-841.

Minor comments

General - mNG is a more common abbreviation for mNeonGreen than mNeG

Line 103-104: delete “activity” in “activating sPG synthesis activity”

Line 154: replace “which is” at the end of the sentence by “and is”

Line 290 – typo in “stationary”

Page 9, SI, below equations – “its mutants”, not “it’s mutants”

Sup Fig 8 – state meaning of CDF in legend.

Thanks for spotting these mistakes. We have corrected all of them in the text.

Reviewer #2 (Remarks to the Author):

The manuscript by Lyu et al. reports on single-molecule studies of FtsN molecules in live *E. coli* cells. It has been proposed in the past that FtsN is the trigger for the onset of constriction in *E. coli*. The authors find that midcell FtsN proteins can be divided between two groups based on their speeds. One group consists of stationary molecules while the other of slow-moving molecules with speeds of about 10 nm/s. The speed of slow-moving FtsN is dependent on the septal peptidoglycan synthesis but not on FtsZ treadmilling. The earlier studies by the same group have shown that FtsW molecules (septal peptidoglycan transglycosylase) also show a population that moves at speeds about 10 nm/s, which are involved in septal peptidoglycan synthesis (septal peptidoglycan track). Furthermore, the same studies also showed a different group of FtsW moving at speeds 20-30 nm/s, which corresponds to the speed of FtsZ treadmilling. Based on these data, the authors propose that FtsN activates septal cell wall synthesis by capturing or retaining FtsWI on the septal peptidoglycan track.

The reported experiments are carefully done and reported findings appear overall solid with few exceptions. However, the claim that FtsN activates/trigger septal peptidoglycan synthesis is vague and not backed up by data. Although the work lacks new evidence that FtsN activates the septal cell wall synthesis, the presented data provide valuable insights on how FtsN is involved in cell division. These findings are of broad interest to people who study bacterial cell division and cell cycle.

Main points of criticism:

1) The main claim that FtsN activates/trigger septal peptidoglycan synthesis is vague and not backed up by data. The authors just show that FtsN likely moves together with the FtsI-FtsW.

More careful wording of the title and conclusion in the abstract (lines 31-34) and elsewhere is warranted.

We have revised the title as: "*FtsN maintains active septal cell wall synthesis by forming a processive complex with the septum-specific peptidoglycan synthase in E. coli*" and revised the abstract accordingly.

2) There is inconsistency in the claims on how FtsN is recruited to the septum. On one hand, the authors claim that FtsN activates sPG synthesis by capturing or retaining FtsWI on the sPG-track (abstract). However, in the main summarizing Figure (Fig. 4C) they show that FtsA recruits FtsN although the authors do not see this group of FtsN in their experiments, presumably because the period to observe this group is too transitory. But if FtsN indeed were the trigger, then this triggering would occur during this transitory period. The latter makes the point that the authors cannot claim that FtsN triggers the septal peptidoglycan synthesis as they are not observing it.

We revised the discussion to be consistent with our observations. See Figure 4 legend and discussion lines 476-492.

3) Why is there a population of stationary FtsN Cyto-TM molecules in Fig. 2J? How is Fig. 2I consistent with the explanation that stationary population corresponds to FtsN that is bound to sPG via its SPOR domain? How is it ruled out the stationary population is not an analysis artifact? See also note to Lines 200-203 below.

As we described above in addressing Review 1's comments, the stationary population of FtsN^{Cyto-TM} is the one associated with FtsA which bound to stationary FtsZ monomers in the middle of treadmilling FtsZ polymers, similar to what we previously observed on FtsWI. See lines 309-315 and 446-451.

4) The summary model in Fig. 4C is not consistent with the observations and previous reports:

1. There is no evidence of FtsN in the Z-track (rightmost part of the Figure). This should be observable in the experiments if, as the authors propose "FtsN is first recruited to the septum through the interaction between its cytoplasmic tail with FtsA, and is distributed around the septum by treadmilling FtsZ polymers." The distribution process should presumably take some time (about 1 minute to traverse half a perimeter).

We did not observe full length FtsN on the Z-track because our data suggest that the periplasmic interactions of FtsN are stronger than the FtsN-FtsA cytoplasmic interaction. However, this transitory population could be deduced based on the observations that FtsN variants missing the SPOR domain showed a fast-moving population on the Z-track. The proposed model is consistent with this reasoning and available genetic data (FtsN is recruited first to the septum by its interaction with FtsA).

2. It is not clear that the stationary population corresponds to FtsN that is bound to sPG via its SPOR domain because FtsN Cyto-TM shows a stationary population.

These two stationary populations are different. The stationary population of FtsN^{Cyto-TM} is most likely the one associated with FtsA which bound to stationary FtsZ monomers in treadmilling FtsZ polymers, while the stationary population of full length FtsN is likely the one bound to denuded glycan chains. The two stationary populations also have different dwell times (~ 30 s vs ~ 10 s for

full length FtsN and FtsN^{Cyto-TM} respectively). Please see lines 309-315 and 446-451 for more details.

3. The current model supported by different groups suggests that one of the players in activating septal peptidoglycan synthesis is FtsQLB as has also been shown by authors in Figure 1A. However, the model in Figure 4C does not involve FtsQLB. None of the results in the manuscript show that FtsQLB complex is involved.

Figure 4C (now Figure 4D) has been redrawn to include FtsQLB. We are currently investigating the dynamics and role of FtsQLB, but this work is beyond the scope of the current manuscript.

Minor points of criticism:

1) Lines 133-134: “However, autocorrelation analysis showed that the FtsN molecules in the FtsN-ring are more homogenously distributed than those in the FtsZ-ring (Fig. 1C)”. Fig.1C shows not an autocorrelation function but a pair-correlation function. The authors should specify if the distance r is a 3D distance or the distance on the cylindrical surface.

Figure 1C is an autocorrelation function using the formula published in one of our previous works (Coltharp *et al.*, PNAS, 2016):

$$\rho(r) = \frac{1}{\sigma^2} \left\{ \lim_{L \rightarrow \infty} \frac{1}{L} \int_0^L Z(x) \cdot Z(x+r) dx \right\}$$

Where the distance r is the linear distance along the circumference of the ring, as we described in Supplementary information lines 208-211.

2) Fig 1B: The radial thickness of the FtsN-ring is listed 51 +/- 4 nm. It is not feasible that the position of mEos3.2 can vary more than a few nanometers from the inner membrane because FtsN is an integral membrane protein. The listed number is rather experimental uncertainty. The same seems to apply also to the thickness of the FtsZ-ring because the numbers are comparable to that of the FtsN-ring.

This is an interesting point. We do not believe this measurement reflects our spatial resolution, because in calculating the width we have already deconvolved out the spatial resolution as we described previously (Coltharp *et al.*, PNAS, 2016). We believe this width may reflect the width of the leading edge of the invaginating, V-shaped membrane that is occupied by FtsN, similar to what was previously proposed in Söderström *et al.*, Mol Microbiol, 2018.

3) Lines 138-141: “We found that FtsN-rings assemble at a ring diameter of ~ 600 nm and disassemble at ~ 300 nm (Fig. 1D). In contrast, under the same experimental condition FtsZ-rings assemble at ~ 950 nm and start to disassemble at ~ 600 nm (Fig. 1D).” These numbers are not consistent. FtsZ- and FtsN rings should assemble at the same diameter if FtsN “activates” septal cell wall synthesis. Also, 950 nm for E. coli diameter is too large.

It is unclear why FtsN appears to assemble at a smaller diameter than FtsZ, but we did not observe any FtsN rings with large ring diameters that are comparable to FtsZ rings in our data. Most likely we did not have enough pre-divisional cells in our samples. Another possible explanation could be that PBP3-independent PG synthesis (PIP) occurs during the ~ 20 min delay between FtsZ and FtsN’s arrivals at the septum, which causes initial cell wall constriction at an early stage.

E. coli's typical diameter is ~ 900 nm to 1 μm under most laboratory growth conditions (Bronk *et al.*, Biophysics J, 1996; El-Hajj *et al.*, Front Microbiol, 2015).

4) Lines 197-198: "The Halo tag is inserted after amino acid E60, between the TM and E domains in the periplasm". Please explain why the Halo tag was inserted into this region instead of the cytoplasmic domain as in the measurements described earlier.

We added text to explain this sandwich fusion. See lines 205-208: "*Here we switched from N-terminal fusions to a sandwich fusion because we could use the same Halo insertion site when comparing the dynamics of full-length FtsN to those of FtsN derivatives that lack the cytoplasmic domain or periplasmic domain.*"

5) Lines 198-199: "We tracked septum-localized single FtsN-HaloSW molecules using a frame rate of 1 Hz to effectively filter out fast, randomly diffusing molecules along the cylindrical part of the cell body." I do not think the frame rate here is a relevant number to quote. One should instead mention the exposure time.

We added the exposure time (100 ms).

6) Lines 200-203: "Using a custom-developed unwrapping algorithm, 47, we decomposed 3D trajectories of individual FtsN molecules obtained from the curved cell surfaces at midcell to one-dimensional (1D) trajectories along the circumference and long axis of the cell respectively as previously described 47" The unwrapping algorithm is very sensitive to accurate determination of cell contours. The authors should explain how cell contours were determined and how the shift between brightfield and fluorescent images was handled. This information is missing from Ref 47.

The unwrapping procedures was described in ref 47 Supplementary Fig. 3 and Methods section "Single-molecule tracking of FtsW-RFP and data analysis: FtsW-RFP tracking".

As control of their method, the authors should plot the speed of FtsN-Halosw versus the radial distance of the molecule from the cell center.

This is indeed a very interesting point. We have done so on FtsW and the results show that the speeds do not change based on the cell diameter, but the population shifts from the fast to the slow population. We decided not to include these data in the currently manuscript but will integrating them in an upcoming manuscript focusing on the progression of cell wall constriction.

7) Fig. 2H: the legend overshadows the data.

We used a smaller legend font size in the new Fig. 2H to avoid this problem.

8) Lines 254-256: "Both mutant fusions were able to support cell division as the sole cellular FtsN and showed prominent midcell localization, but cells were both longer than WT ones (Fig. 2F and Supplementary Fig. 10)". Please mention in the text how much longer.

Revised to include this information ("*about twice as long as WT*"). See line 269.

9) Figure 4A: Are FtsN WT cells shown here already with superfission variant ftsBE56A ΔftsN background? If yes, then add this information to the Figure legend.

Yes. FtsN^{WT} was put into the *ftsB*^{E56A} Δ *ftsN* background. We now included this information in the figure caption.

10) Lines 391-393: “In other words, the SPOR domain may be the major determinant to prevent the release of the sPG synthesis complex from the sPG-track to the Z-track.” Is there any model that proposes that sPG synthesis complex is released from the sPG-track to the Z-track? What would be the function of such a release?

In the two-track model, the release of the sPG synthesis complex from sPG-track to the Z-track constitutes the inactivation process. Conversely, the release of the sPG synthesis complex from the Z track to the sPG track constitute the activation process. The switching back and forth between the two tracks controls the enzymes' activities.

11) Lines 401-402: “In principle FtsN might localize to the Z-track and prevent or even disrupt binding of FtsWI to the Z-track.” How is this statement consistent with the conclusion from the authors' previous paper that “FtsN promotes FtsW release from the Z-track to become active in sPG synthesis on the slow 'sPG-track'”?

The previous statement that “FtsN promotes FtsW release from the Z-track to become active in sPG synthesis on the slow 'sPG-track' is consistent with the current statement that “In principle FtsN might localize to the Z-track and prevent or even disrupt binding of FtsWI to the Z-track”, because FtsN could stay on the Z-track to drive away FtsW from Z-track, in other words, promotes FtsW's release from the Z-track.

12) Line 447:” we found that abrogating the FtsN-FtsA interaction, either by deleting the cytoplasmic domain or introducing a D5N substitution, resulted in mild cell elongation”. This is misleading according to data shown on SI Fig.10, which shows that cell length increase 2x and 4x.

We have edited the text to focus on the consequences for FtsN dynamics and removed any mention of elongation, as this is not critical to the point of the experiment. The revised sentence now reads: “*In support of this notion, we found that abrogating the FtsN-FtsA interaction, either by deleting the cytoplasmic domain or introducing a D5N substitution, did not diminish the slow-moving population of FtsN molecules on the sPG-track (Fig. 2F-H).*” (Lines 478-481). The cell length data are still in Fig. S11 (formerly S10) for readers who are interested.

13) Lines 457-460: “In this model, FtsN is first recruited to the septum through the interaction between its cytoplasmic tail with FtsA, and is distributed around the septum by treadmilling FtsZ polymers. This period may be too transitory for us to observe a significant population of fast-moving, full length FtsN molecules in our experiments.” If FtsN is appreciably spread over the perimeter of the cell by treadmilling then at speed 25 nm/s and should be observable. Some further explanation is warranted.

We do not know how long the spreading period needs to be to distribute FtsN molecules throughout the septum. We are currently working on a computational model to gain insight into this process.

Reviewer #3 (Remarks to the Author):

The manuscript by Lyu et al. studies the dynamic behavior of the essential cell division protein FtsN in *E. coli*. FtsN has at three distinct domains: an N-terminal cytoplasmic peptide that interacts

with FtsA, the essential E-domain in the periplasm that is known to interact with and activate FtsWI and the SPOR-domain that binds to denuded peptidoglycan strands. While it seems clear that all these interactions are involved in the recruitment of FtsN towards the division septum, it's not known how they individually contribute to control the timing of recruitment starting from the initial assembly of the Z-ring to the constriction of the cell septum. From previous work we already know the interaction with the PG synthesis machinery is dominating during constriction, as there is a clear spatial separation between the FtsN and FtsZ. In contrast, FtsN remained colocalized with FtsI (Söderström et al Molecular Microbiology 2018). This study also found that the movements of FtsZ and FtsN are mechanistically different as demonstrated by their spatial separation and different fluorescence recoveries in FRAP experiments. Together, this study concluded that the FtsZ and FtsN are part of two distinct protein complexes that are becoming spatially separated during constriction. The idea of separated complexes was picked up by another recent study from the group of Jie Xiao (Yang et al. Nature Microbiology 2021). Here, single molecule imaging helped to identify two tracks that gave rise to different dynamics of the septal cell wall synthesis complex FtsWI: first, a fast Z-track of inactive enzymes whose movement is driven by FtsZ treadmilling dynamics and second, a slow PG-track, where proteins move slowly powered by PG synthesis. This study found that FtsN activates FtsW and promotes its switch from fast to slow motion. From these two studies, it is already clear that FtsN is first recruited to the septum via an interaction with FtsA but then forms a complex with FtsWI and PG. In contrast, there is no existing evidence for an alternative model, where FtsN would remain part of the Z-track during the entire process of cell division.

In the current paper, the authors analyze the single molecule behavior of FtsN. Consistent with the study by Söderström et al and Yang et al, they find that FtsN shows two different populations, an immobile and a moving one. In the later population, FtsN moves at a similar velocity as FtsWI suggesting that its dynamics are not powered by FtsZ treadmilling, but by PG synthesis. This again confirms a model where FtsN predominantly localizes to the PG synthesis machinery. Next, the authors construct a couple of different FtsN truncations and find that the dynamics of the N-terminal peptide are defined by its interaction with the Z-ring and that the interaction with FtsWI can drive the dynamics of the E-domain. These are interesting observations but again confirm already existing studies (Busiek et al 2012, 2014, Yahashiri et al 2017).

Overall, I think that studying the intracellular dynamics of cell division proteins is a powerful approach that offers new mechanistic insight, however, this paper does not provide any new information regarding the function and properties of FtsN.

We respectfully disagree with the reviewer. Although FtsN has been regarded as the sPG synthesis activator, how it does so is unclear. We provided new experimental observations that septal FtsN molecules move processively at $\sim 9 \text{ nm s}^{-1}$, the same as FtsWI molecules engaged in sPG synthesis (termed sPG-track), but much slower than the $\sim 30 \text{ nm s}^{-1}$ speed of inactive FtsWI molecules coupled to FtsZ's treadmilling dynamics (termed FtsZ-track). Importantly, processive movement of FtsN is exclusively coupled to sPG synthesis and is required to maintain active sPG synthesis by FtsWI. Our findings indicate that FtsN is part of the FtsWI sPG synthesis complex, and that while FtsN is often described as a "trigger" for the initiation for cell wall constriction, it must remain part of the processive FtsWI complex to maintain sPG synthesis activity. All these observations are new and provide new insight into the question of how FtsN activate sPG synthesis. We also document for the first time that there are at least two stationary FtsN subpopulations: one that stays bound to denuded glycan and one that is bound to internal FtsZ monomers; the FtsZ-bound population is observed only if FtsN's SPOR domain is deleted, indicating the SPOR domain is important not only for septal localization but for which track FtsN

localizes to at the septum (Z-track or sPG-track). Overall, we think the manuscript provides considerable new information.

My biggest disappointment is that single molecule imaging of FtsN should in principle provide a wealth of data about its behavior during cell division but the authors do not take advantage of it. For example, from the velocity histograms of single molecule trajectories it is obvious that FtsNs exist in different complexes. From the single molecule trajectories, it should be possible to obtain information about the lifetime of these complexes and the respective switching rates between them. These numbers should change during cell division, i.e. be significantly different from the early stages of Z-ring assembly towards cell separation and should be very different for the different truncations. This kind of information is already would be incredibly useful to reveal a much more detailed picture at how FtsN is recruited to midcell and when and how it then moves on to activate FtsWI.

We want to thank the reviewer for his/her deep appreciation for the capability of single-molecule tracking. The reviewer is absolutely correct that single molecule tracking offers rich information for identifying transition kinetics between different subcomplexes. From the data we have in this manuscript, we document that there are at least three FtsN subpopulations: one that stays bound to denuded glycans, which has a lifetime of ~ 30-40 s, another that is bound to FtsWI and engaged in processive movement of sPG synthesis (processive lifetime of ~ 15 s), and a third stationary subpopulation in the absence of the SPOR domain that is bound to internal FtsZ monomers with a lifetime of ~ 10-15 s.

We are very excited by the prospect of identifying transition kinetics between different FtsN subcomplexes, and are currently collecting more data in a different format to obtain longer and more complete trajectories along the circumference of the septum for these kinetic analyses. However, we feel that including these analyses is beyond the scope of the current work, as the manuscript already documents the existence of the three subpopulations of FtsN. The manuscript also characterizes one of them in detail, namely, the subpopulation in a complex with FtsWI. We show that (a) its formation is mediated by FtsN's E domain, that (b) its movement is powered by ongoing sPG synthesis, that (c) its movement is independent of FtsZ's treadmilling activity, and that (d) FtsN must remain part of the complex to keep it active rather than interacting with FtsWI only transiently as suggested by the frequent description of FtsN as a "trigger". Performing a similarly detailed characterization of the other FtsN subcomplexes, as suggested by Reviewer 3, would likely triple the length of the paper and require the collection of hundreds if not thousands more long trajectories in a different format to tease out transition kinetics and pinpoint individual subpopulations with statistical rigor. Our current datasets are robust enough to document the existence of the other subcomplexes but not enough to identify transition kinetics. This task also requires analyzing FtsN dynamics in some new mutant backgrounds, including, but not limited to, mutants that lack cell wall hydrolases to perturb the lifetime of denuded glycan strands to which FtsN's SPOR domain binds. For these reasons, we believe that completing and writing up these detailed analyses belong in a new manuscript instead of the current one, even though we completely agree with the Reviewer that these FtsN subpopulations are worthy of detailed analyses.

In addition, it is disappointing that the manuscript does not include any single molecule data (time lapse micrographs) that could support their findings and also include previously published data without properly disclosing it.

We provided supplementary movies and all single-molecule tracking trajectories will be released upon publication.

In summary, due to the lack of new observations, missing analysis and data, I cannot recommend publication of the manuscript.

Please see our rebuttal above.

Please find my more specific comments below:

Fig. 1:

This figure does not show new information, instead it is a confirmation of findings by Söderström et al. 2018. This should be properly discussed.

We did provide new information—please see lines 130-148. Briefly, we showed that the FtsN-ring has similar dimensions compared to the Z-ring, that it assembles and disassembles at a much later stage during cell wall constriction than the FtsZ-ring, and that it has a much slower FRAP recovery rate than the Z-ring. Importantly, FRAP reveal a significant population of FtsN that remains stationary on the time scale of the experiment, which we later show is that this population is most likely bound to denuded glycan chains. We also show that at most 20% of FtsN is in the FtsN ring, which has implications for the number of active FtsWI complexes and their stoichiometries.

- Panel B contains data from a previously published manuscript, this should be clearly disclosed in the main text and figure caption.

We added this information in the figure caption.

- N-terminal mEos3.2-FtsN is missing in Supplementary Table 4. FtsN localization is not confined to midcell (compare with immunostainings in supplement). Immunostaining for mEos3.2 FtsN to confirm correct localization

We added a new figure (see below, now is Supplementary Fig. 3) to characterize the mEos3.2-FtsN fusion. mEos3.2-FtsN showed correct midcell localization as expected. We added this information in Supplementary Table 4.

- Can the authors do a significance test to demonstrate the difference between the autocorrelation profile? To me they do look similar. If the authors believe this analysis is useful, it should be done at different time points during division and for different versions of FtsN, i.e. is it still different for FtsN Cyto TM? What about FtsN delta Cyto TM?

The two ACF curves are significantly different as indicated by the associated error bars.

We do not believe comparing the ACF of mutant FtsN is necessary, as it is clear from the following SMT experiments that the dynamics, instead of the structure of FtsN-rings, is more important for FtsN's function.

- Panel C and D is a less detailed characterization of the spatial separation of FtsZ and FtsN observed before, it does not include new information.

See the point above—these measurements were done with a higher spatial resolution and hence provide better dimension measurement and ACF analysis.

- Incomplete FRAP recovery also due to limited amount of FtsN molecules in the living it is no indication of a distinct FtsN species. The supporting movie is not convincing.

We normalized the recovery percentage to the other side of the ring (unbleached area), which takes into account the photobleaching and the limited total number of FtsN molecules in the cell.

Fig. 2:

- What is the lifetime of stationary complexes? Denuded Glycans are only transiently present in SPG, this could be measured by getting a histogram of stationary lifetimes.

We have measured the stationary lifetime. They are listed in Supplementary Tables 7-10. In general, the lifetime is 30-40 s.

- Line 199: 1Hz framerate is too slow to visualize fast diffusive behavior and short-lived states, which would be interesting to study to understand the behavior of FtsN. It is also incorrect that a long frame rate filters out random diffusion, long exposure time does this.

In this work we did not focus on fast diffusing FtsN molecules as we demonstrated that processively moving FtsN molecules are important for FtsWI's activity. We did observe fast, freely diffusing FtsN molecules in and out of the septum, which will be part of a new manuscript.

We revised the text to include the long exposure time (100 ms).

- Where are there time lapse movies of single molecule experiments?

We have now added three supplementary movies 3-5 showing processive movement of FtsN molecules.

- Please use better color coding, some plots are difficult to understand (Fig. 2H and 2K for example)

These two plots are composites from different samples and we wish to distinguish each sample differently. We used a smaller legend font size in the new Fig. 2H and added a new legend in Fig. 2K. Hope it is easier to understand now.

- What can explain the immobile fraction of the different FtsN truncations? While it is straightforward to explain this for the full-length protein, it's not clear how for FtsN Cyto TM.

Please refer to our responses to the stationary populations in Reviewers 1 and 2.

- Line 290: "The rest FtsN segments..." something is missing in this sentence

It is a typo. We have deleted this sentence.

- Instead of using FtsN D5N, could the authors use an N-terminal truncation?

We did. See Figure 2F, FtsN^{ΔCyto-TM}.

Fig. 3:

- What explains the stationary protein?

Please see above.

- Or is it possible to de- or increase the presence of denuded peptidoglycan to see if this stationary fraction is depending on the denuded PG?

Yes, and we are currently conducting these experiments, which will be included in a future manuscript.

Reviewer Comments, second round

Reviewer #1 (Remarks to the Author):

The manuscript by Lyu and colleagues has been considerably improved, both by the additional experiments made, as well as by the clear explanations now introduced in the text that clarify some relevant points, such as the existence of stationary molecules of FtsN variants that lacked the SPOR domain. I have no further comments except the one below.

Reviewer 2 made a relevant point saying that the fact that FtsN ring assembled at a ring diameter of ~600nm was not consistent with the established idea that FtsN activates septal cell wall synthesis. Given that the authors do not have a clear explanation for this, this inconsistency should be mentioned either in the results or in the discussion.

Reviewer #2 (Remarks to the Author):

The revised manuscript by Lyu et al. has improved by more accurately summarizing the findings. However, several significant questions about the results and their interpretation still remain. In particular, the explanation that stationary population FtsN can be explained by FtsN binding to FtsA appears not consistent with FtsNCyto-TM data.

The authors also explain in their responses that "[...]. We decided not to include these data in the current manuscript but will integrate them in an upcoming manuscript focusing on the progression of cell wall constriction." The authors should include these data in the current manuscript. The data in the manuscript are not trustworthy without this control.

Below are my follow-up comments to the responses from the authors. Comments from the previous review are numbered, the authors' responses start with > and my follow up comments by >>

Main points of criticism:

3) Why is there a population of stationary FtsN Cyto-TM molecules in Fig. 2J? How is Fig. 2I consistent with the explanation that the stationary population corresponds to FtsN that is bound to sPG via its SPOR domain? How is it ruled out the stationary population is not an analysis artifact? See also note to Lines 200-203 below.

> As we described above in addressing Review 1's comments, the stationary population of FtsNCyto-TM is the one associated with FtsA which bound to stationary FtsZ monomers in the middle of treadmilling FtsZ polymers, similar to what we previously observed on FtsWI. See lines 309-315 and 446-451.

>> How comes then that there is also a moving population of FtsNCyto-TM? What would explain this movement and that its speed matches FtsZ treadmilling speed? If FtsNCyto-TM binds to FtsA, which monomers are stationary then FtsNCyto-TM population should be also all stationary. The explanation does not add up.

1. Since the lifetimes of the stationary population were introduced, the authors should mention in the manuscript what is the photobleaching lifetime of fluorophores.
2. Also, how do the stationary FtsN molecules behave after the stationary period? Would they disappear or start moving?

4) The summary model in Fig. 4C is not consistent with the observations and previous reports:

There is no evidence of FtsN in the Z-track (rightmost part of the Figure). This should be observable in the experiments if, as the authors propose "FtsN is first recruited to the septum

through the interaction between its cytoplasmic tail with FtsA, and is distributed around the septum by treadmilling FtsZ polymers." The distribution process should presumably take some time (about 1 minute to traverse half a perimeter).

> We did not observe full length FtsN on the Z-track because our data suggest that the periplasmic interactions of FtsN are stronger than the FtsN-FtsA cytoplasmic interaction. However, this transitory population could be deduced based on the observations that FtsN variants missing the SPOR domain showed a fast-moving population on the Z-track. The proposed model is consistent with this reasoning and available genetic data (FtsN is recruited first to the septum by its interaction with FtsA).

>> I do not still see evidence in the author's data on FtsN being part of treadmilling FtsZ protofilaments (left side of the schematics). Also based on the earlier data X. Yang et al. NatMicrobiol 2021 FtsIW should be part of the fast track (left side of the schematics).

Minor points of criticism:

1) Lines 133-134: "However, autocorrelation analysis showed that the FtsN molecules in the FtsN-ring are more homogeneously distributed than those in the FtsZ-ring (Fig. 1C)". Fig. 1C shows not an autocorrelation function but a pair-correlation function. The authors should specify if the distance r is a 3D distance or the distance on the cylindrical surface.

> Figure 1C is an autocorrelation function using the formula published in one of our previous works (Coltharp et al., PNAS, 2016):

$$\rho(r) = 1 \left\{ \lim_{L \rightarrow \infty} \frac{1}{L} \int_0^L Z(x) \cdot Z(x+r) dx \right\}^2$$

Where the distance r is the linear distance along the circumference of the ring, as we described in Supplementary information lines 208-211.

>> Please add the formula to SI and explain the notions. What is Z ? L going to infinity in this formula does not make sense because the circumference of the ring is finite.

2) Fig 1B: The radial thickness of the FtsN-ring is listed 51 +/- 4 nm. It is not feasible that the position of mEos3.2 can vary more than a few nanometers from the inner membrane because FtsN is an integral membrane protein. The listed number is rather experimental uncertainty. The same seems to apply also to the thickness of the FtsZ-ring because the numbers are comparable to that of the FtsN-ring.

> This is an interesting point. We do not believe this measurement reflects our spatial resolution, because in calculating the width we have already deconvolved out the spatial resolution as we described previously (Coltharp et al., PNAS, 2016). We believe this width may reflect the width of the leading edge of the invaginating, V-shaped membrane that is occupied by FtsN, similar to what was previously proposed in Söderström et al., Mol Microbiol, 2018.

>> The authors should then plot the thickness of the ring versus the ring radius. For larger radii the thickness should be much smaller; otherwise, the explanation is without substance.

3) Lines 138-141: "We found that FtsN-rings assemble at a ring diameter of ~ 600 nm and disassemble at ~ 300 nm (Fig. 1D). In contrast, under the same experimental condition FtsZ-rings assemble at ~ 950 nm and start to disassemble at ~ 600 nm (Fig. 1D)." These numbers are not consistent. FtsZ- and FtsN rings should assemble at the same diameter if FtsN "activates" septal cell wall synthesis. Also, 950 nm for E. coli diameter is too large.

> It is unclear why FtsN appears to assemble at a smaller diameter than FtsZ, but we did not observe any FtsN rings with large ring diameters that are comparable to FtsZ rings in our data. Most likely we did not have enough pre-divisional cells in our samples. Another possible explanation could be that PBP3-independent PG synthesis (PIP) occurs during the ~ 20 min delay between FtsZ and FtsN's arrivals at the septum, which causes initial cell wall constriction at an early stage.

>> What valuable information then the reader learns from this discussion? These numbers then just reflect measurement bias.

6) Lines 200-203: "Using a custom-developed unwrapping algorithm, 47, we decomposed 3D trajectories of individual FtsN molecules obtained from the curved cell surfaces at midcell to one-dimensional (1D) trajectories along the circumference and long axis of the cell respectively as previously described 47" The unwrapping algorithm is very sensitive to accurate determination of cell contours. The authors should explain how cell contours were determined and how the shift between brightfield and fluorescent images was handled. This information is missing from Ref 47.

> The unwrapping procedures was described in ref 47 Supplementary Fig. 3 and Methods section "Single-molecule tracking of FtsW-RFP and data analysis: FtsW-RFP tracking".

>> Sorry, it is not adequately described in ref 47. Again, the unwrapping algorithm is very sensitive to the accurate determination of cell contours and there should be some controls to show that this has been properly performed.

As control of their method, the authors should plot the speed of FtsN-Halosw versus the radial distance of the molecule from the cell center.

> This is indeed a very interesting point. We have done so on FtsW and the results show that the speeds do not change based on the cell diameter, but the population shifts from the fast to the slow population. We decided not to include these data in the current manuscript but will integrate them in an upcoming manuscript focusing on the progression of cell wall constriction.

>> The authors should show these data in the current manuscript. Your data are not trustworthy without this control.

12) Line 447: "we found that abrogating the FtsN-FtsA interaction, either by deleting the cytoplasmic domain or introducing a D5N substitution, resulted in mild cell elongation". This is misleading according to data shown on SI Fig.10, which shows that cell length increase 2x and 4x.
> We have edited the text to focus on the consequences for FtsN dynamics and removed any mention of elongation, as this is not critical to the point of the experiment. The revised sentence now reads: "In support of this notion, we found that abrogating the FtsN-FtsA interaction, either by deleting the cytoplasmic domain or introducing a D5N substitution, did not diminish the slow-moving population of FtsN molecules on the sPG-track (Fig. 2F-H)." (Lines 478-481). The cell length data are still in Fig. S11 (formerly S10) for readers who are interested.

>> The information about elongation is still highly relevant and should be mentioned.

13) Lines 457-460: "In this model, FtsN is first recruited to the septum through the interaction between its cytoplasmic tail with FtsA, and is distributed around the septum by treadmilling FtsZ polymers. This period may be too transitory for us to observe a significant population of fast-moving, full length FtsN molecules in our experiments." If FtsN is appreciably spread over the perimeter of the cell by treadmilling then at speed 25 nm/s and should be observable. Some further explanation is warranted.

> We do not know how long the spreading period needs to be to distribute FtsN molecules throughout the septum. We are currently working on a computational model to gain insight into this process.

>> According to new explanations in the manuscript, if FtsA is stationary and FtsN binds to this stationary FtsA then how would this distribution mechanism operate?

Additional comments to the newly edited text:

Lines 435-436: We further determined that FtsN is present at ~300 molecules per cell under our growth conditions.

I do not see any evidence that the authors' determined this number. The number comes from Li&Weissman 2014.

Reviewer #3 (Remarks to the Author):

I want to thank the authors for the improved manuscript and the clarifications. I agree with the conclusions and think the paper can be published.

We thank the reviewers for their constructive comments. Below we provide point-to-point responses (blue) to reviewers' comments (black). To facilitate the reading of the revised manuscript, we highlighted major changes in blue in both the main text and supplemental information. Small typos and grammar corrections are not highlighted.

Reviewer #1 (Remarks to the Author):

The manuscript by Lyu and colleagues has been considerably improved, both by the additional experiments made, as well as by the clear explanations now introduced in the text that clarify some relevant points, such as the existence of stationary molecules of FtsN variants that lacked the SPOR domain. I have no further comments except the one below.

Thank you!

Reviewer 2 made a relevant point saying that the fact that FtsN ring assembled at a ring diameter of ~600nm was not consistent with the established idea that FtsN activates septal cell wall synthesis. Given that the authors do not have a clear explanation for this, this inconsistency should be mentioned either in the results or in the discussion.

We now provide more in-depth discussions of why we observed that FtsN rings assemble at a smaller septum diameter (~ 600 nm) than FtsZ rings. We believe that the small amount of FtsN molecules recruited to the septum at the onset of constriction may not be high enough to create the appearance of the ring. It is possible that only when sufficient denuded glycans accumulate in the nascent division septum, a larger amount of FtsN molecules is recruited to make the appearance of the ring. Our data suggest that happens at round 600 nm. We have added that explanation in lines 436-443:

"...We observed that FtsN forms a discontinuous or patchy ring-like structure and exhibits distinct septal organization and dynamics compared to those of the Z-ring. FtsN-rings were first visible as such at a septal diameter of ~ 600 nm while FtsZ-rings at ~ 950 nm, The difference in their timing of ring assembly could reflect the fact that the small amounts of FtsN recruited at the onset of constriction do not create the appearance of a ring, which only occurs after sufficient denuded glycans accumulate in the nascent division septum to recruit a larger amount of FtsN. Thus, the 600 nm diameter may reflect the transition from primarily FtsA-mediated to primarily denuded glycan-mediated FtsN localization¹⁷."

Reviewer #2 (Remarks to the Author):

The revised manuscript by Lyu et al. has improved by more accurately summarizing the findings. However, several significant questions about the results and their interpretation still remain. In particular, the explanation that stationary population FtsN can be explained by FtsN binding to FtsA appears not consistent with FtsNCyto-TM data.

We thank the reviewer's affirmation of our improvements. There are two FtsZ- associated populations of FtsN^{Cyto-TM}, one is stationary, associated with stationary FtsZ monomers in the middle of treadmilling FtsZ polymers, and the other is directionally moving by end-tracking the shrinking end of FtsZ polymers through a Brownian ratchet mechanism, as we previously showed for FtsWI (McCausland et al., Nat. Commun., 2021; Yang et al.,

Nat. Microbiol., 2021). Both populations are likely mediated through the binding of FtsN to FtsA, which binds to FtsZ.

The authors also explain in their responses that “[...]. We decided not to include these data in the current manuscript but will integrate them in an upcoming manuscript focusing on the progression of cell wall constriction.” The authors should include these data in the current manuscript. The data in the manuscript are not trustworthy without this control.

We believe the data we presented in this manuscript support our conclusions on FtsN's dynamics. The constriction-dependent behavior requested here is not a control experiment for the current manuscript, but a full investigation that warrants its own manuscript.

Below are my follow-up comments to the responses from the authors. Comments from the previous review are numbered, the authors' responses start with > and my follow up comments by>>

Main points of criticism:

3) Why is there a population of stationary FtsN Cyto-TM molecules in Fig. 2J? How is Fig. 2I consistent with the explanation that the stationary population corresponds to FtsN that is bound to sPG via its SPOR domain? How is it ruled out the stationary population is not an analysis artifact? See also note to Lines 200-203 below.

> As we described above in addressing Review 1's comments, the stationary population of FtsNCyto-TM is the one associated with FtsA which bound to stationary FtsZ monomers in the middle of treadmilling FtsZ polymers, similar to what we previously observed on FtsWI. See lines 309-315 and 446-451.

>> How comes then that there is also a moving population of FtsNCyto-TM? What would explain this movement and that its speed matches FtsZ treadmilling speed? If FtsNCyto-TM binds to FtsA, which monomers are stationary then FtsNCyto-TM population should be also all stationary. The explanation does not add up.

As noted above, there are two populations of FtsN^{Cyto-TM} associated with FtsZ polymers. The stationary population is linked to internal positions in FtsZ polymers. The moving population of FtsN^{Cyto-TM} is end-tracking the shrinking end of treadmilling FtsZ polymers as we explained previously. The Brownian ratchet mechanism is able to bias the diffusion of FtsN^{Cyto-TM} to end-track continuously the shrinking end of treadmilling FtsZ polymers, hence matching their speeds. When a shrinking end of an FtsZ polymer meets stationary FtsN-FtsA molecules bound in the middle of the FtsZ polymer, these molecules may diffuse away stochastically or start end-tracking the new shrinking end of FtsZ polymers. We refer the reviewers to our previous publications on the detailed mechanism of these behaviors (*McCausland et al., Nat. Commun.*, 2021; *Yang et al., Nat. Microbiol.*, 2021).

1. Since the lifetimes of the stationary population were introduced, the authors should mention in the manuscript what is the photobleaching lifetime of fluorophores.

The lifetime of stationary population in fixed cells is 73.7 ± 2.3 s (Supplementary Table 10), which represents the photobleaching lifetime of fluorophores under the same imaging condition. The fluorophore lifetime is much longer than all the lifetimes of the stationary

population under different conditions (Supplementary Tables 7-10, 12), indicating the observed stationary population lifetimes were not limited by the fluorophore lifetime.

2. Also, how do the stationary FtsN molecules behave after the stationary period? Would they disappear or start moving?

Most of them started moving (see the representative trajectory in Figure 2C, where a stationary FtsN molecule started to move at ~ 40 s). Others disappeared either because they disassociated from the septum and diffused away or they were photobleached (see the representative trajectory in Figure 2A, where a stationary FtsN molecule disappeared at ~ 100 s).

4) The summary model in Fig. 4C is not consistent with the observations and previous reports:

There is no evidence of FtsN in the Z-track (rightmost part of the Figure). This should be observable in the experiments if, as the authors propose “FtsN is first recruited to the septum through the interaction between its cytoplasmic tail with FtsA, and is distributed around the septum by treadmilling FtsZ polymers.” The distribution process should presumably take some time (about 1 minute to traverse half a perimeter).

> We did not observe full length FtsN on the Z-track because our data suggest that the periplasmic interactions of FtsN are stronger than the FtsN-FtsA cytoplasmic interaction. However, this transitory population could be deduced based on the observations that FtsN variants missing the SPOR domain showed a fast-moving population on the Z-track. The proposed model is consistent with this reasoning and available genetic data (FtsN is recruited first to the septum by its interaction with FtsA).

>> I do not still see evidence in the author’s data on FtsN being part of treadmilling FtsZ protofilaments (left side of the schematics). Also based on the earlier data X. Yang et al. NatMicrobiol 2021 FtsIW should be part of the fast track (left side of the schematics).

There is evidence for FtsN in the Z-track, namely, the behavior of FtsN^{Cyto-TM}, for which we clearly document a Z-track population. As explained in the discussion (lines 491-498), we consider this derivative of FtsN to mimic the behavior of full-length FtsN early in the constriction process when denuded glycans are not yet abundant enough to draw FtsN away from the Z-track. If Reviewer #2 has an alternative explanation that accounts for the data, we’d like to hear it.

As for the expectation that full-length FtsN should be seen on the Z-track if it is “first recruited” by FtsA, we now recognize that our wording here may have been misinterpreted and have revised the legend of Figure 4 to be clearer (lines 860-873). By “first recruited” we meant at the onset of constriction, before denuded glycans are present in appreciable amounts. Later in constriction, essentially 100% of FtsN becomes available to FtsWI complexes via release from sPG. It was not our intention to say that FtsA is also the entry pathway for FtsN in constricted cells, though we now appreciate our wording seems to suggest that.

Minor points of criticism:

1) Lines 133-134: "However, autocorrelation analysis showed that the FtsN molecules in the FtsN-ring are more homogeneously distributed than those in the FtsZ-ring (Fig. 1C)". Fig.1C shows not an autocorrelation function but a pair-correlation function. The authors should specify if the distance r is a 3D distance or the distance on the cylindrical surface.

> Figure 1C is an autocorrelation function using the formula published in one of our previous works (Coltharp et al., PNAS, 2016):

$$\rho(r) = \frac{1}{\sigma^2} \left\{ \lim_{L \rightarrow \infty} \frac{1}{L} \int_0^L Z(x) \cdot Z(x+r) dx \right\}$$

Where the distance r is the linear distance along the circumference of the ring, as we described in Supplementary information lines 208-211.

>> Please add the formula to SI and explain the notions. What is Z ? L going to infinity in this formula does not make sense because the circumference of the ring is finite.

Z is the number of molecules along the circumference of the ring when the ring is unwrapped to become a one-dimensional line for the calculation. The equation is the mathematical definition of autocorrelation function of a theoretically infinitely long trajectory. L going to infinity means to take the limit of the length (or time) average. We refer the reviewer to the book "An introduction to stochastic processes with applications to Biology" by Linda Allen for some background information. In practice, all trajectories are finite, and hence the theoretical equation takes the formula from below:

$$p(r_k) = \frac{\sum_{i=1}^{N-k} Z_i Z_{i+k}}{\sum_{i=1}^N Z_i^2} \quad k = 0, 1, \dots, N-1$$

where i is the index of the trajectory ($Z_1, Z_2, Z_3 \dots Z_N$), k is the distance lag between individual data points, and N is the total number of data points in a trajectory.

We have added the information in the Supplemental Information (lines 206-211).

2) Fig 1B: The radial thickness of the FtsN-ring is listed 51±4 nm. It is not feasible that the position of mEos3.2 can vary more than a few nanometers from the inner membrane because FtsN is an integral membrane protein. The listed number is rather experimental uncertainty. The same seems to apply also to the thickness of the FtsZ-ring because the numbers are comparable to that of the FtsN-ring.

>This is an interesting point. We do not believe this measurement reflects our spatial resolution, because in calculating the width we have already deconvolved out the spatial resolution as we described previously (Coltharp et al., PNAS, 2016). We believe this width may reflect the width of the leading edge of the invaginating, V-shaped membrane that is occupied by FtsN, similar to what was previously proposed in Söderström et al., Mol Microbiol, 2018.

>> The authors should then plot the thickness of the ring versus the ring radius. For larger radii the thickness should be much smaller; otherwise, the explanation is without substance.

We plotted the thickness of the FtsN-ring versus the ring diameter as shown below. (Measurements from individual cells were plotted as small gray circles at respective ring diameters, and the averaged measurements in different bins were plotted as large red circles. Error bars represent *s.e.m.*). The N-ring thickness remained essentially constant throughout constriction, indicating that there was no significant structural reorganization of the N-ring during constriction. It was similar as FtsZ-ring in our previous work (Lyu *et al.*, *Biopolymers*, 2016). There is no reason to expect why the thickness of the ring to decrease as constriction progresses.

3) Lines 138-141: “We found that FtsN-rings assemble at a ring diameter of ~ 600 nm and disassemble at ~ 300 nm (Fig. 1D). In contrast, under the same experimental condition FtsZ-rings assemble at ~ 950 nm and start to disassemble at ~ 600 nm (Fig. 1D).” These numbers are not consistent. FtsZ- and FtsN rings should assemble at the same diameter if FtsN “activates” septal cell wall synthesis. Also, 950 nm for *E. coli* diameter is too large.

> It is unclear why FtsN appears to assemble at a smaller diameter than FtsZ, but we did not observe any FtsN rings with large ring diameters that are comparable to FtsZ rings in our data. Most likely we did not have enough pre-divisional cells in our samples. Another possible explanation could be that PBP3-independent PG synthesis (PIP) occurs during the ~ 20 min delay between FtsZ and FtsN’s arrivals at the septum, which causes initial cell wall constriction at an early stage.

>> What valuable information then the reader learns from this discussion? These numbers then just reflect measurement bias.

See the explanation at the very beginning of addressing Reviewer #2’s comments. We thank the reviewer for prompting us to think harder about this issue and now offer an alternative explanation related to the time needed to accumulate denuded glycans and thus sufficient FtsN to form what looks like a ring. By this logic, the number reflect a biologically meaningful step in the division process, not simply an artifact of under-sampling.

6) Lines 200-203: “Using a custom-developed unwrapping algorithm, 47, we decomposed 3D trajectories of individual FtsN molecules obtained from the curved cell surfaces at midcell to one-dimensional (1D) trajectories along the circumference and long axis of the cell respectively as previously described 47” The unwrapping algorithm is very sensitive to accurate determination of cell contours. The authors should explain how cell contours were determined and how the shift between brightfield and fluorescent images was handled. This information is missing from Ref 47.

> The unwrapping procedures was described in ref 47 Supplementary Fig. 3 and Methods section “Single-molecule tracking of FtsW-RFP and data analysis: FtsW-RFP tracking”.

>> Sorry, it is not adequately described in ref 47. Again, the unwrapping algorithm is very sensitive to the accurate determination of cell contours and there should be some controls to show that this has been properly performed.

We believe it is clearly described in ref 47 Supplementary Fig. 3 and Methods section “Single-molecule tracking of FtsW-RFP and data analysis: FtsW-RFP tracking”. Fig. 3 and its legend explain how the cell contour is determined and calibrated using both bright field and superresolution images (as shown below).

This figure describes the custom-developed cell envelope unwrapping method to retrieve true coordinates of single molecules along the circumference of the curved cell surface. **a.** The apparent cell diameter ($D_{\text{bright field}}$, right panel) is measured using the intersections of the background signal (red line) with the cell profile (black curve) resulted from a line scan (yellow dashed line) of the cell's bright-field image (left panel). Scale bar: 1 μm . **b.** For each cell (BW25113/pJL005), the corresponding diameter of FtsZ ring ($D_{3\text{dPALM}}$) is measured using three-dimensional (3D) superresolution image of FtsZ-mEos3.2 (left) fit with a circle (right). Scale bar: 0.5 μm . **a and b.** Micrographs are representative of independent experiments. **c.** The apparent cell diameter $D_{\text{bright field}}$ is plotted against $D_{3\text{dPALM}}$ from the same cell and fit with a line with a slope of 1 and intersection at 57nm (magenta line). **a, b, and c:** $n = 52$ cells with Z-ring. **d.** The true cell radius (R , from cell center to the inner membrane) is calculated from $D_{\text{bright field}}$ subtracting the intersection as calculated in **c** and adding the distance between FtsZ ring and inner membrane (~ 17 nm). The 'true' coordinate X_{real} (purple arc) of a molecule along the circumference of the cell inner membrane is calculated using the true cell radius R and the detected X coordinate X_{detect} (bottom equation). **e.** One segment of a single FtsW-RFP trajectory along the circumference fits to a line. The displacement (L) and the segment length (T) introduced in the Methods are labeled in the top panel. The noise level S is defined as the standard deviation of the residuals (s_i , bottom panel).

As control of their method, the authors should plot the speed of FtsN-Halosw versus the radial distance of the molecule from the cell center.

> This is indeed a very interesting point. We have done so on FtsW and the results show that the speeds do not change based on the cell diameter, but the population shifts from the fast to the slow population. We decided not to include these data in the current manuscript but will integrate them in an upcoming manuscript focusing on the progression of cell wall constriction.

>> The authors should show these data in the current manuscript. Your data are not trustworthy without this control.

See the explanation in the beginning.

12) Line 447: "we found that abrogating the FtsN-FtsA interaction, either by deleting the cytoplasmic domain or introducing a D5N substitution, resulted in mild cell elongation". This is misleading according to data shown on SI Fig.10, which shows that cell length increase 2x and 4x.

> We have edited the text to focus on the consequences for FtsN dynamics and removed any mention of elongation, as this is not critical to the point of the experiment. The revised sentence now reads: "In support of this notion, we found that abrogating the FtsN-FtsA interaction, either by deleting the cytoplasmic domain or introducing a D5N substitution, did not diminish the slow- moving population of FtsN molecules on the sPG-track (Fig. 2F-H)." (Lines 478-481). The cell length data are still in Fig. S11 (formerly S10) for readers who are interested.

>> The information about elongation is still highly relevant and should be mentioned.

We measured the cell length (Figure S11).

13) Lines 457-460: "In this model, FtsN is first recruited to the septum through the interaction between its cytoplasmic tail with FtsA, and is distributed around the septum by treadmilling FtsZ polymers. This period may be too transitory for us to observe a significant population of fast- moving, full length FtsN molecules in our experiments." If FtsN is appreciably spread over the perimeter of the cell by treadmilling then at speed 25 nm/s and should be observable. Some further explanation is warranted.

> We do not know how long the spreading period needs to be to distribute FtsN molecules throughout the septum. We are currently working on a computational model to gain insight into this process.

>> According to new explanations in the manuscript, if FtsA is stationary and FtsN binds to this stationary FtsA then how would this distribution mechanism operate?

Only FtsA-FtsN complexes that are bound to the middle of FtsZ polymers are stationary. Others that are end-tracking FtsZ polymers and are distributed by treadmilling.

Additional comments to the newly edited text:

Lines 435-436: We further determined that FtsN is present at ~300 molecules per cell under our growth conditions.

I do not see any evidence that the authors' determined this number. The number comes from Li&Weissman 2014.

Actually, the passage in question referenced a supplemental figure with a quantitative Western (Figure S5). We have now revised the sentence to spell that out. Our estimate from Westerns agrees very well with the prior estimate from Li *et al.* based on RNA profiling.

Reviewer #3 (Remarks to the Author):

I want to thank the authors for the improved manuscript and the clarifications. I agree with the conclusions and think the paper can be published.

Thank you!

Reviewer Comments, third round

Reviewer #2 (Remarks to the Author):

The authors in their responses have offered some convincing explanations such as how to explain the stationary FtsN-Cyto-TM population yet neglected the question about showing a control of their cell envelope unwrapping procedure. The authors state that "We believe the data we presented in this manuscript support our conclusions on FtsN's dynamics. The constriction-dependent behavior requested here is not a control experiment for the current manuscript, but a full investigation that warrants its own manuscript." I have not been asking for any new measurements but for a plot based on the author's existing data. The plot that I'd like to see to be convinced about the accuracy of their method is $X_{\text{(detect)}}/R$ vs V for wild-type cells. The notations follow ref 47 Supplementary Fig. 3, which is also included in their rebuttal letter. The authors could pool together data from different cells with different R and show all the data without binning. All these data exist and making the plot is trivial. Without seeing this data, I have rather serious concerns about their method. As $X_{\text{(detect)}}$ approaches R , all the molecules appear stationary and the accuracy of determining V becomes zero.

I do not agree with the explanation that "There is evidence for FtsN in the Z-track, namely, the behavior of FtsNCyto-TM, for which we clearly document a Z-track population. As explained in the discussion (lines 491-498), we consider this derivative of FtsN to mimic the behavior of full-length FtsN early in the constriction process when denuded glycans are not yet abundant enough to draw FtsN away from the Z-track. If Reviewer #2 has an alternative explanation that accounts for the data, we'd like to hear it." There is no evidence that FtsNCyto-TM mimics the behavior of full-length FtsN early in the constriction process. For some reason, the authors do not show any data of full-length FtsN early in the constriction process. As such, much cannot be concluded, but it is a prerogative of the authors to interpret their results in the Discussion based on their views. However, if pitfalls are known these should be mentioned. One of the alternative interpretations is that there is no FtsN in the Z-track.

Finally, the added explanation to the manuscript "...We observed that FtsN forms a discontinuous or patchy ring-like structure and exhibits distinct septal organization and dynamics compared to those of the Z-ring. FtsN-rings were first visible as such at a septal diameter of ~ 600 nm while FtsZ-rings at ~ 950 nm, The difference in their timing of ring assembly could reflect the fact that the small amounts of FtsN recruited at the onset of constriction do not create the appearance of a ring, which only occurs after sufficient denuded glycans accumulate in the nascent division septum to recruit a larger amount of FtsN. Thus, the 600 nm diameter may reflect the transition from primarily FtsA-mediated to primarily denuded glycan-mediated FtsN localization¹⁷". I do not follow why "the appearance of a ring" is important. The authors presumably track individual molecules. ~ 600 nm diameter is pertinent to how the authors analyze their data. It is not clear what role FtsA plays in the implied transition.

Reviewer #2 (Remarks to the Author):

The authors in their responses have offered some convincing explanations such as how to explain the stationary FtsN-Cyto-TM population yet neglected the question about showing a control of their cell envelope unwrapping procedure. The authors state that “We believe the data we presented in this manuscript support our conclusions on FtsN’s dynamics. The constriction-dependent behavior requested here is not a control experiment for the current manuscript, but a full investigation that warrants its own manuscript.” I have not been asking for any new measurements but for a plot based on the author’s existing data. The plot that I’d like to see to be convinced about the accuracy of their method is X_{detect}/R vs V for wild-type cells. The notations follow ref 47 Supplementary Fig. 3, which is also included in their rebuttal letter. The authors could pool together data from different cells with different R and show all the data without binning. All these data exist and making the plot is trivial. Without seeing this data, I have rather serious concerns about their method. As X_{detect} approaches R , all the molecules appear stationary and the accuracy of determining V becomes zero.

The reviewer misunderstood our method. As shown in the Figure A below, the focal plane (marked as green) was placed at ~ 200 nm from the bottom of the cell to image molecules moving on the bottom half of the cylindrical portion of the cell body (see more details in the Methods Section in the main text of *Yang et al., Nat. Microbiol., 2021*). We selected this imaging plane so that molecules in the side areas (marked as red) would not be in focus and hence not detected. Additionally, we purposely avoid molecules toward the edge of the arc in order to minimize the error in identifying their speeds. Therefore, X_{detect} would rarely approach R .

The reviewer thought “All these data exist and making the plot is trivial”, which is not the case. A single trajectory was segmented into multiple short ones, with each short segment having a corresponding speed V (see Figure 2C in the main text). We do not calculate X_{detect} but only use X_{real} after the unwrapping procedure to calculate V . Therefore, each cell that has an R will have multiple V segments. In Figure B below we plotted D (cell diameter) vs V (all the speeds including moving and stationary segments). It does not show any dependence between D and V .

I do not agree with the explanation that “There is evidence for FtsN in the Z-track, namely, the behavior of FtsNCyto-TM, for which we clearly document a Z-track population. As explained in the discussion (lines 491-498), we consider this derivative of FtsN to mimic the behavior of full-length FtsN early in the constriction process when denuded glycans are not yet abundant enough

to draw FtsN away from the Z-track. If Reviewer #2 has an alternative explanation that accounts for the data, we'd like to hear it." There is no evidence that FtsNCyto-TM mimics the behavior of full-length FtsN early in the constriction process.

We believe that FtsN^{Cyto-TM} mimics the behavior of full length of FtsN at the early constriction stage because only the Cyto-TM domain of FtsN is required for FtsN's early septum localization. Furthermore, the functions of the periplasmic domains (SPOR and E) of FtsN are not required yet at this stage because not many denuded glycan strands are available for FtsN's SPOR domain to bind and sPG synthesis has also not fully commenced to require the binding between FtsN's E domain and FtsWI. These reasonings are built upon a large body of past literature showing that the isolated N-terminal fragment of FtsN provides biologically meaningful information on the behavior of the full-length protein, including early localization (*Busiek et al., J Bacteriol., 2012; Busiek et al., Mol. Microbiol., 2014; Liu et al., Mol. Microbiol., 2015; Söderström et al., Mol. Microbiol., 2016; Baranova et al., Nat. Microbiol., 2020; Radler et al., Nat. Commun., 2022*). It is important to note that it is a common practice in biology to use mutants to reveal phenomena not visible in WT.

For some reason, the authors do not show any data of full-length FtsN early in the constriction process.

We did not purposely "not show" or exclude any full-length FtsN data early in the constriction process. We just simply did not see them in statistically sufficient numbers to determine their significance. We further offer the following possible reasons for why we did not see them.

First, as shown by our data (Figure 1B, D) and many other groups' work (one example of demographs from *Verheul et al., PloS Genet., 2022* is shown on the right), there are very few FtsN molecules in cells in the early constriction stage. Therefore, there is less of a chance for us to see FtsN molecules in the early constriction stage.

Second, the early constriction process could be very short so that only a small population of cells are in that stage, further decreasing the chance to capture full-length FtsN molecules at the midcell in the early constriction stage.

Third, in SMT experiments, we needed to use a very low concentration of JF646 dye (1nM) to sparsely label the FtsN-Halo^{SW} fusion so that no more than one FtsN-Halo^{SW} molecule was labeled per cell, which resulted some cells not being labeled at all.

Fourth, FtsN's dynamics could be too transient to track in the early constriction stage. In our analysis, we filtered out trajectories shorter than 5 frames (which is 5 s) to exclude randomly diffusing molecules in the septum and to minimize classification error (see more details in the SMT imaging and data analysis Section in SI). Even though we observed many longer trajectories in the Cyto-TM mutant, it's possible that we filtered out some shorter trajectories that are "real" but too transient in both Cyto-TM mutant and full-length FtsN.

As such, much cannot be concluded, but it is a prerogative of the authors to interpret their results in the Discussion based on their views. However, if pitfalls are known these should be mentioned.

The major pitfalls in our study, as in any other single-molecule studies, are the stochasticity in molecules' behaviors and the detection limit. It is challenging to capture rare events, which requires scaling up experiments to a level sometimes not practically feasible. We are also limited by the fluorophore's photoproperties such as brightness and photostability.

One of the alternative interpretations is that there is no FtsN in the Z-track.

This alternative is possible to someone who may not be familiar with the field, but it is against a large body of past literature, as the interactions between FtsA and FtsN are well documented. Nevertheless, we now address the topic directly in the revised manuscript (line 280-283 and 321-323).

Finally, the added explanation to the manuscript "...We observed that FtsN forms a discontinuous or patchy ring-like structure and exhibits distinct septal organization and dynamics compared to those of the Z-ring. FtsN-rings were first visible as such at a septal diameter of ~ 600 nm while FtsZ-rings at ~ 950 nm, The difference in their timing of ring assembly could reflect the fact that the small amounts of FtsN recruited at the onset of constriction do not create the appearance of a ring, which only occurs after sufficient denuded glycans accumulate in the nascent division septum to recruit a larger amount of FtsN. Thus, the 600 nm diameter may reflect the transition from primarily FtsA-mediated to primarily denuded glycan-mediated FtsN localization¹⁷".

I do not follow why "the appearance of a ring" is important. The authors presumably track individual molecules. ~600 nm diameter is pertinent to how the authors analyze their data.

The reviewer likely mixed up two different experiments, Single-molecule localization based superresolution microscopy (SMLM) and Single-molecule tracking (SMT). Scoring midcell localization as a ring as shown in Figure 1D was done in the SMLM experiment, where we found FtsN-rings were first visible at a septal diameter of ~ 600 nm. In order to make out the shape of a ring, one needs to have a sufficient number of data points along the ring. See the illustration below. If there are only two molecules (red dots) being localized along the ring (gray), it is impossible to make out the shape of a ring. As the number of data points increases, the ring shape will emerge. This effect is formally termed the Nyquist criterion.

In SMT experiments, we didn't exclude any cells in the analysis according to their diameters. We looked at the trajectories of labeled FtsN in every single cell without any bias, then only filtered out trajectories outside the midcell or shorter than 5 frames. It is true that the SMT assay could track a single molecule (even in the early constricted cells without an apparent ring), but that one molecule does not allow us to score it as forming a ring.

Finally, the field has long noted that there is a lag between FtsZ localization and FtsN localization. In those studies, localization of both proteins was defined by the appearance of a ring. We use the same criterion but now relate that to cell diameter. Our conclusion that FtsN localizes later than FtsZ and that the FtsN ring is patchy are not dependent on the 600 nm diameter.

It is not clear what role FtsA plays in the implied transition.

It is well established in the field that FtsN is recruited by at least two interactions, first with FtsA and later with denuded glycans. FtsA-recruited FtsN activates FtsWI, which coincides with the activation of Amidases. Amidases generate denuded glycan to recruit more FtsN to the septum, allowing the transition into the more active cell wall constriction phase.

Reviewer Comments, fourth round

Reviewer #2 (Remarks to the Author):

The authors have addressed the concerns that I have brought up. However, they should mention in the methods section that the focal plane was ~ 200 nm below the cell bottom in their imaging experiments as they explain in their rebuttal letter.

Reviewer #2 (Remarks to the Author):

The authors have addressed the concerns that I have brought up. However, they should mention in the methods section that the focal plane was ~200 nm below the cell bottom in their imaging experiments as they explain in their rebuttal letter.

We did mention it in the Methods section. It's in line 812 in the revised manuscript. Actually, we placed the focal plane at ~ 250 nm above the bottom of the cell in the real experiment.